# Intrinsic Entropy of Context Length Scaling in LLMs

**Jingzhe Shi**[1,3*], **Qinwei Ma**[1,3*], **Hongyi Liu**[2*], **Hang Zhao**[1^], **Jeng Neng Hwang**[4], **Lei Li**[4^]

[1]Institute for Interdisciplinary Information Sciences, Tsinghua University
[2]Carnegie Mellon University, [3]CPHOS,[†] [4]University of Washington

[1]sjzworking@gmail.com,{mqw21@mails,hangzhao@mail}.tsinghua.edu.cn,
[2]hongyil2@andrew.cmu.edu, [3]hwang@uw.edu, lenny.lilei.cs@gmail.com,
[*]Equal Contribution [^]Equal Correspondence

## Abstract

Long Context Language Models have drawn great attention in the past few years. There has been work discussing the impact of long context on Language Model performance: some find that long irrelevant context could harm performance, while some experimentally summarize loss reduction by relevant long context as Scaling Laws. This calls for a more thorough understanding of how long context impacts Language Modeling. In this work, we (1) propose to use 'Intrinsic Entropy' for explaining the impact of context length on language modeling; and (2) conduct experiments on natural language and synthetic data, validating our proposed theoretical assumptions and deductions. Our theoretical framework can provide practical insights such as establishing that training dataset size dictates an optimal context length and bounds context length scaling for certain cases. We hope our work may inspire new long context Language Models, as well as future work studying the physics of Language Models.[1]

## 1 Introduction

As language-model capacity has rapidly increased and long context has become crucial for tasks such as reasoning and retrieval, recent work has focused on extending context length. A variety of methods have been proposed to support long-context language models (Su et al., 2023; Katharopoulos et al., 2020; Gu & Dao, 2024; Peng et al., 2023; Sun et al., 2024). At the same time, prior work reports mixed outcomes: some studies show that long irrelevant context worsens LM performance (Xu et al., 2024; Levy et al., 2024), some summarize gains from relevant long context as scaling laws (Xiong et al., 2024), and work in other domains such as time series shows that even relevant long context can hurt performance (Shi et al., 2024). **These observations call for a more thorough understanding of how context length affects language-model performance.**

Previous theories explain scaling laws with respect to dataset and model size (Bahri et al., 2024; Sharma & Kaplan, 2020; Chen et al., 2025). However, most of them do not study how context length impacts scaling laws for language modeling, thus they cannot contribute directly to the problem.[2]

In this work, we analyze the impact of context length by decomposing the total loss into 2 components. As shown in Figure 1, the components are: the **Bayes Risk**, representing the loss of an optimal language model given certain context length, and the **Approximation Loss**, the loss caused by the gap between the optimal language model and the trained model. The total loss is the sum of these 2 components, the balance between which would **possibly** lead to seemingly counter-intuitive behaviors like *longer context worsens the performance*. Such analysis works for analyzing impact of context length on multiple scenarios, such as for pretraining targeting Cross-Entropy loss, or when evaluating down-stream metrics, as shown in Figure 1.

---

[†]CPHOS is an academic non-profit organization.
[1]Code for experiments is available at: https://github.com/JingzheShi/NLPCtlScalingAndBounds.
[2]We discuss more about previous work in Appendix J.

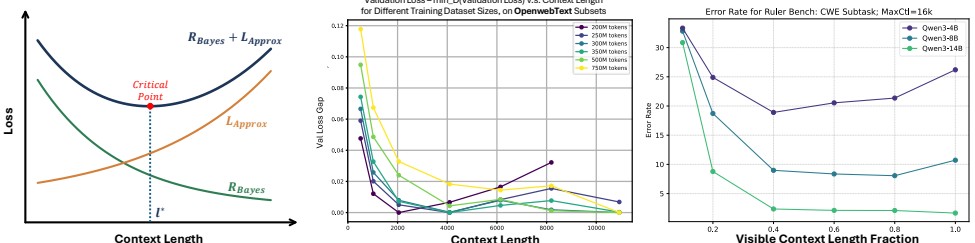

Figure 1: **Left**: Total loss is decomposed into Bayes Risk (decreasing with context length) and Approximation Loss (increasing with context length), so a critical point can emerge in some scenarios. **Middle**: Validation Loss Gap (Val Loss - $\min_D$(Val Loss) vs. Context Length), measured on subsets of OpenWebText, where we subtract the minimum loss within each dataset-size curve (please refer to Figure 15 for the original figure). For each training dataset size, there exists an optimal context length that minimizes pretraining validation loss, and this optimum increases with dataset size. **Right**: Error rate of Qwen series models on the CWE task from RulerBench when a certain amount of context is masked. Critical points are visible. More results on other RulerBench subsets are shown in Figure 8.

Building on this decomposition, we introduce **Intrinsic Entropy**, which measures how much information is available to an LLM at a given context length for a dataset. Starting from simple assumptions about Intrinsic Space and information entropy, we derive the relationship among **Cross Entropy Loss, Intrinsic Entropy, and Context Length**. We then validate these assumptions and deductions on both natural language and synthetic data. **Our main contributions are**:

- We analyze context-length effects through the trade-off between **Bayes Risk** and **Approximation Loss**, showing that longer context does not always improve performance.
- We introduce **Intrinsic Entropy** and use it to explain language modeling behavior across different context lengths.[3]
- We validate the theoretical assumptions and deductions with experiments on both real language data and synthetic data.

We hope our work may inspire future work when it comes to explaining context impact and/or training new long context Language Models.

## 2 ASSUMPTIONS, DEDUCTIONS AND OBSERVATIONS FOR LANGUAGE MODELING

### 2.1 PRELIMINARIES

#### 2.1.1 PRELIMINARY: LOSS DECOMPOSITION

It is common in ML studies to decompose the loss into **Bayes Risk** (the minimum loss possible, achieved by the theoretically optimal Bayesian Model), and **Approximation Loss** (the loss measuring the ability of a trained model actually to approximate the Bayesian Model). Specifically for Cross-Entropy loss $H$, we have (please refer to **Appendix B.1 for formal definitions and derivation details**):

$$\begin{aligned} H(P, Q_l) =&R_{Bayes} + L_{Approx} \\ =&H(P, P_l) + D_{KL}(P_l, Q_l) \end{aligned} \tag{1}$$

Where $P = p(x_0|x_{-\infty:0})$ is the distribution of Natural Language (or our experimented dataset), $P_l = p(x_0|x_{-l:0})$ is the Bayesian Model for context length $l$ and $Q_l = q(x_0|x_{-l:0})$ is the

---

[3] 'Intrinsic Space', the basis of 'Intrinsic Entropy', is commonly defined as the middle-layer feature representation of well-trained neural networks, and we follow this practice in the main paper. In Appendix D, we also provide formal definitions for these assumptions.

learned Language Model of context length $l$. $R_{Bayes} = H(P, P_l)$ is the **Bayes Risk** of optimal model (the assumed 'limit' when we have infinite data points and model parameters) and $L_{Approx} = D_{KL}(P_l, Q_l)$ is the **Approximation Loss**, which can be affected by dataset size $D$, etc. The Bayes Risk is model or data agnostic, only related to natural language itself and is limited only by visible context length.

### 2.1.2 PRELIMINARY: INTRINSIC SPACE

In previous work (Bahri et al., 2024; Cheng et al., 2023), as a common practice, the 'Data Manifold' is often **defined** as the middle feature representation of well-trained neural networks, and **assumptions** are made on this kind of mid-representation, with experiments to **validate** these assumptions. (Intrinsic Space is defined as the space where the Data Manifold lies.) We follow such practice in main paper for clarity.

Meanwhile, the Data Manifold can be more formally defined by a mapping from input data to some Intrinsic Space which satisfies a certain set of properties, and mid-representation of well-trained neural networks are assumed to have such properties, which can be experimentally validated. This is an equivalent yet more formal perspective. In Appendix D, we formally define the Intrinsic Space and derive related results in our work with such perspective for completeness.

### 2.1.3 PRELIMINARY: OUTLINES

In **Section 2.2** we propose the definition of Intrinsic Entropy, and discuss how to bridge **Bayes Risk** with it, thus explaining how context length impacts Bayes Risk.

Approximation Loss, or how well the trained model learns Bayesian Model, is related to Intrinsic Dimension in previous work of Scaling Laws (Sharma & Kaplan, 2022; Shi et al., 2024). In **Section 2.3** we discuss more about how the context length impacts **Approximation Loss** from this perspective.

We further derive that the balance between **Bayes Risk** and **Approximation Loss** would lead to an optimal context length which increases with the size of the training dataset. Our theoretical deduction and experiments on language are presented in **Section 3**.

## 2.2 BAYES RISK WITH CONTEXT LENGTH: AN INTRINSIC ENTROPY PERSPECTIVE

In this section we discuss to bridge context length and Bayes Risk with the concept of Intrinsic Entropy.

### 2.2.1 BAYES RISK AND ENTROPY IN INTRINSIC SPACE: DERIVED FROM FIRST PRINCIPLES

'Information Entropy' is defined as the amount of information carried in the Intrinsic Space. Here are detailed assumptions[4] as definitions: [5]

- **Assumption 1**. Information Entropy of Intrinsic Space for Bayes Model $\lim_{l\to\infty} S(P_l) = S(P_\infty)$ is finite, which is the Information Entropy of next token prediction of language itself.

- **Assumption 2**. $\forall l_1, l_2$ such that $l_1 < l_2$, $S(P_{l_1}) < S(P_{l_2})$. This is because a longer context contains more information.

- **Assumption 3**. **Linear Entropy Relationship**: The Information Entropy w.r.t. Next Token Prediction, defined as $S_{ntp}(P_l) = H(P_0) - H(P_l)$, is linear with the Entropy in the Intrinsic Space of the Bayes Model, i.e., $S_{ntp}(P_l) = k * S(P_l) + b$, and $0 < k < 1$. **A formal definition can be found in Appendix D**.

  $S_{ntp}$ is smaller than $S$ since the Intrinsic Space contains important information on previous tokens that are important for the prediction of future tokens, while $S_{ntp}$ is related only to the next token. The hidden state in RNNs contain more information than only the next token to predict. For example, consider a character-level RNN that predicts the sentence '1 + 2

---

[4]Appendix D defines Intrinsic Entropy from a more formal perspective.
[5]To avoid confusion, we use 'H' for 'Cross Entropy Loss', and 'S' for 'Information Entropy'.

equal_', the next character to predict is 's', but the hidden state should contain information about answer '3' for the latter tokens.

With these assumptions, we can derive that the Bayes Risk is linear with respect to the Intrinsic Entropy:

$$R_{Bayes} = H(P, P_l) \\ = -k * S(P_l) + Const \tag{2}$$

This **linear relationship** is observed in experiments for LMs in **Section 2.2.2**, and for synthetic data in **Section 4.3**.

Note that by Assumptions 1 and 2 we derive: [6] $\frac{\partial R_{Bayes}}{\partial l} < 0$, and $\lim_{l \to \infty} \frac{\partial R_{Bayes}}{\partial l} = 0$.

### 2.2.2 BAYES RISK AND INTRINSIC ENTROPY: EXPERIMENT MEASUREMENT

We use well-trained Large Language Models to conduct experiments for approximating the Bayes Risk $H(P_l)$ on certain text corpora. We find that:

$$H(P, P_l) \approx C_0 + C/l^\gamma \tag{3}$$

approximates the experimented behavior well on both OpenWebText and other text corpora (please refer to Appendix H and Appendix E for detailed figures and results on different datasets).

**Experimentally measure Intrinsic Entropy using Gaussian-KDE**   To validate the linear relationship between Bayes Risk and Intrinsic Entropy (Equation 2), we measure the Intrinsic Entropy in the hidden representation space of well-trained Language Models. For a given context length, we gather the hidden state of the final layer for the last token across multiple ($\geq 10000$) samples, and use Gaussian Kernel Density Estimation (Gaussian-KDE) to estimate the Information Entropy of the distribution in this space. We also provide an alternative eigenvalue-based estimation method in Appendix H.

We conduct experiments on multiple Language Models: Llama-3.1-8B, Qwen3-8B-Base, and RecurrentGemma-9B, all evaluated on a subset of the OpenWebText dataset. As shown in Figure 2, the linear relationship between Cross Entropy loss and Gaussian-KDE measured Intrinsic Entropy holds across different model architectures, validating our theoretical assumptions:

$$R_{Bayes} \approx -k * S(P_l) + Const,$$

which aligns well with **Equation 2**, thus validating our entropy-based deduction. Note that for RecurrentGemma-9B, several outlier points at very low context lengths exhibit significantly higher CE loss than other models, indicating that RecurrentGemma-9B is not a good approximation of the Bayes Model at those context lengths; excluding these outliers, the linear relationship still holds.

### 2.3 APPROXIMATION LOSS WITH CONTEXT LENGTH: AN INTRINSIC DIMENSION PERSPECTIVE

**Approximation Loss in the Training Scenario**   Previous work summarizes scaling laws (Kaplan et al., 2020; Hoffmann et al., 2022) as $L_{Approx}(D) = C_0 + A/D^\alpha$ for dataset size $D$. This has been explained from an intrinsic-space perspective in work such as (Bahri et al., 2024; Sharma & Kaplan, 2022), where $\alpha \approx c/dim$ and $dim$ is the manifold dimension of the data/model under a uniform-distribution assumption in intrinsic space. **In Appendix D.2, we derive this rigorously from weaker assumptions (Theorem 1, Theorem 2).**

---

[6]Generally speaking, the context length $l$ is an integer. Here, following previous work (Kaplan et al., 2020; Tao et al., 2024), we assume $R_{Bayes}$ admits a differentiable extension $\tilde{R}_{Bayes}$ to real-valued $l$ and use $\partial R_{Bayes}/\partial l$ to denote $\partial \tilde{R}_{Bayes}(l)/\partial l$ evaluated at integer $l$. In this sense, the derivative serves as a continuous approximation to the discrete difference $R_{Bayes}(l+1) - R_{Bayes}(l)$. We use the same convention for expressions of the form $df(l)/dl$ throughout this work.

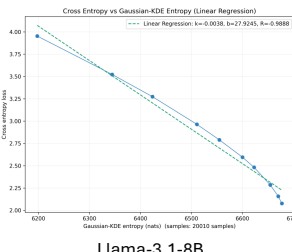 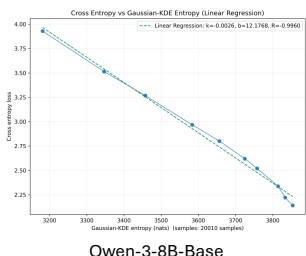 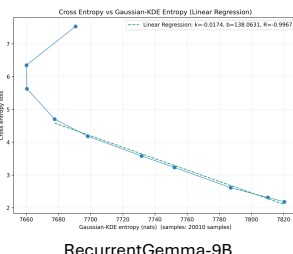

| Llama-3.1-8B | Qwen-3-8B-Base | RecurrentGemma-9B |

Figure 2: Cross Entropy loss vs. Gaussian-KDE measured Intrinsic Entropy (in nats) for three Language Models on a subset of OpenWebText: Llama-3.1-8B (left, $k = -0.0038$, $R = -0.9888$), Qwen3-8B-Base (middle, $k = -0.0026$, $R = -0.9960$), and RecurrentGemma-9B (right, $k = -0.0174$, $R = -0.9967$, with 3 outlier points at high CE loss excluded from regression). The linear relationship between CE loss and Intrinsic Entropy holds across different model architectures.

As assumed in Section 2.2.1, the Intrinsic Dimension should increase with $l$. Combined with previous results on $\alpha = c/dim(l)$, we have,

$$L_{Approx} = C_0 + A(l)/D^{\alpha(l)},$$
$$\frac{\partial \alpha}{\partial l} < 0. \tag{4}$$

This shows that longer context makes it harder for the model to approximate the Bayes model.

**Approximation Loss in the Inference Scenario**  The analysis above considers the training scenario, where both training data size $D$ and context length $l$ jointly determine approximation loss. We now consider a second case: a pre-trained model with fixed parameters evaluated on downstream tasks with varying *visible* context length $l_{vis}$ at inference time.

In this scenario, model parameters are fixed, and approximation loss $L_{Approx}(l_{vis})$ depends on how well the fixed model approximates the Bayes model $P_{l_{vis}}$ for each visible context length. As $l_{vis}$ increases, $P_{l_{vis}}$ lies in a higher-dimensional intrinsic space, making it harder for a fixed-capacity model to approximate. Hence, $\partial L_{Approx}/\partial l_{vis} > 0$: approximation loss increases with visible context length for a fixed model.

Moreover, for a harder downstream task (e.g., one that requires information spread across a wider context range), the Bayes model is more complex and thus harder to approximate, leading to a larger approximation loss overall. This implies that, for a fixed model, harder tasks have larger approximation loss at any given $l_{vis}$.

## 3 DEDUCTION: OPTIMAL CONTEXT LENGTH

In this section, we present deductions from the theory in Section 2. In both training and inference scenarios, Bayes Risk decreases with context length $l$ (Section 2.2), while Approximation Loss increases with $l$ (Section 2.3). The balance between these two opposing trends leads to an optimal context length.

In general, let $l$ denote context length (either training context length or visible context length at inference), let $\theta_t$ denote task-specific parameters that affect Bayes Risk (e.g., task difficulty $\gamma$ in Position-Weighted Ruler-QA1, or the distribution of relevant information across context), and let $\theta_m$ denote model/data-specific parameters that affect Approximation Loss (e.g., training dataset size $D$, or model capacity). The total loss can be written as:

$$\text{Loss}(l, \theta_t, \theta_m) = R_{Bayes}(l, \theta_t) + L_{Approx}(l, \theta_m), \tag{5}$$

where $\partial R_{Bayes}/\partial l < 0$ with $\lim_{l \to \infty} \partial R_{Bayes}/\partial l = 0$ (Section 2.2), and $\partial L_{Approx}/\partial l > 0$ (Section 2.3). Since $R_{Bayes}$ is a decreasing convex function of $l$ and $L_{Approx}$ is increasing in $l$, the derivative of total loss, $\partial_l \text{Loss} = \partial_l R_{Bayes} + \partial_l L_{Approx}$, transitions from negative (Bayes

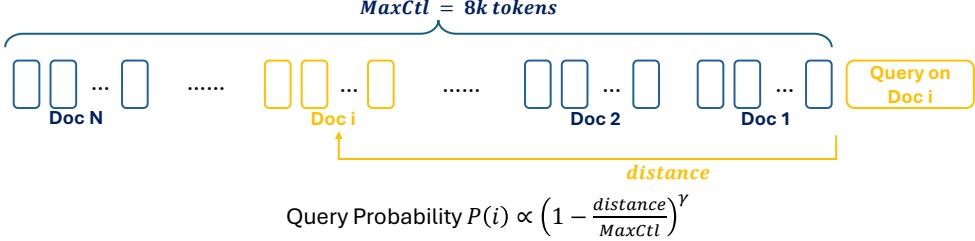

Figure 3: Our modified Position-Weighted Ruler-QA1 (Hsieh et al., 2024) dataset. Multiple paragraphs are concatenated together with context length close to $MaxCtl$, and a question queries the 'golden paragraph' (i.e. the doc paragraph with answer to that query). In the original Ruler-QA1 dataset each doc has equal probability of being queried (i.e. $\gamma = 0$); while in our experiments shown in Figure 4, we measure LLM performance on a set of tasks with different hyper-parameter $\gamma$, each with different probability of querying far-away contexts.

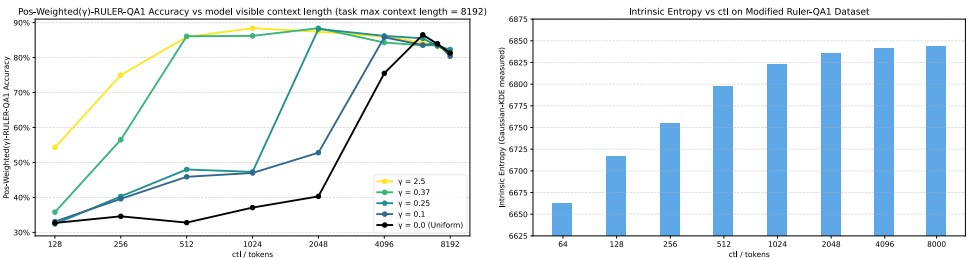

Figure 4: Measured results on Position-Weighted Ruler-QA1 dataset. **Left:** QA accuracy vs. number of tokens input to the Language Model, for different tasks with different $\gamma$ values. We observe that: (1) each curve shows a trend to increase and then decrease with context length; and (2) the critic point corresponds to a smaller optimal context length for tasks with larger $\gamma$ (i.e. tasks requiring less long context abilities). **Right:** Intrinsic Entropy measured on samples truncated to certain context lengths. The Intrinsic Entropy shows increment of intrinsic information when increasing context length, and resembles acc-ctl curves for larger $\gamma$.

Risk dominates) to positive (Approximation Loss dominates), yielding an optimal context length $l^*$ where $\partial_l \text{Loss} = 0$. Note that $R_{Bayes}$ depends on $\theta_t$: in inference, different tasks (e.g., different $\gamma$ in Position-Weighted Ruler-QA1) have different distributions of relevant information across context, leading to different rates at which $R_{Bayes}$ decreases with $l$.

Moreover, when $\theta_m$ varies such that $L_{Approx}$ decreases (e.g., more training data in the training scenario, or a stronger model in the inference scenario), the point where $\partial_l L_{Approx}$ balances $|\partial_l R_{Bayes}|$ shifts to a larger $l$, so the optimal context length $l^*$ increases. Conversely, when $\theta_t$ varies such that $R_{Bayes}$ decreases faster with $l$ (e.g., tasks where relevant information is distributed across a wider context range, corresponding to smaller $\gamma$), Bayes Risk keeps decreasing at larger $l$, also leading to a larger $l^*$.

## 3.1 EXPERIMENTAL MEASUREMENT OF OPTIMAL CONTEXT LENGTH FOR TRAINING

We conduct experiments on a subset of OpenWebText with a sufficiently long context length. We use nanogpt (Karpathy, 2022) and train a model with GPT-2 (Radford et al., 2019) architecture (GPT-2-124M, 12-head transformers, 768-dim feature vector, with half the transformer layers ($12 \rightarrow 6$) to reduce GPU memory for long contexts). We train GPT-2 on different context lengths with different amounts of training data ($200M, 250M, 300M, 350M, 500M, 750M$ tokens), until the validation loss increases.

We show our results in **Figure 1** and **Figure 15**. As shown both theoretically and experimentally, there does exist an optimal context length, **beyond which even relevant long context would increase validation loss** of pretraining Language Models. Such optimal context length would increase with training dataset size. We also provide similar experiments to prove an optimal context length exists on a synthetic dataset, as shown in Appendix F. More details for our experiment settings are presented in Appendix I.

## 3.2 EXPERIMENTAL MEASUREMENT OF OPTIMAL CONTEXT LENGTH ON DOWNSTREAM TASKS

The analysis above focuses on training. We also study context-length effects on downstream tasks, where a trained model is evaluated with varying visible context length. We observe that optimal context length also exists for downstream tasks, and that the optimum increases with task context-length requirements. As shown in Figure 8, on the RULER benchmark, most Qwen3 series models show an optimal context length for qa_1, fwe, and cwe subtasks.

To study the impact of task properties on this 'optimal context length' phenomenon, we propose a Position-Weighted Ruler-QA1 benchmark: instead of a uniform query distribution, query probability depends on the distance of the golden paragraph to the end of input: $P(x) \propto (1 - x/L)^\gamma$. Different $\gamma$ values correspond to tasks focused on different context ranges. As shown in Figure 3 and Figure 4, an optimal context length exists for each $\gamma$, and a smaller $\gamma$ (i.e., a task requiring more long-context ability) typically leads to a larger optimal context length.

This result can be interpreted through Bayes Risk and Approximation Loss decomposition. Intuitively, some tasks require larger context lengths to solve (i.e., the Bayes Risk for that metric decreases more slowly with context length than for metrics such as next-token Cross Entropy), so they benefit from more context. However, because model performance eventually degrades at long context (i.e., Approximation Loss still increases with context length), the balance of these two terms still produces an optimal context length. More details and additional results on models and RULER subtasks are provided in Appendix A.

## 4 PROOF OF CONCEPT WITH SYNTHETIC DATA

### 4.1 LIST OF POINTS TO PROVE

In this section, we conduct experiments on a synthetic dataset, explaining the Bayes Risk and related theories we proposed in Section 2.2. With this synthetic dataset, we would like to prove the following,

- **Point 1**. **Cross Entropy** Loss is approximately linear with **Intrinsic Entropy** (Assumption 3 in Section 2.2.1). Shown in **Section** 4.3.
- **Point 2**. By measuring **Entropy** in **Intrinsic Space** of well-trained models, one could obtain **a valid measurement that is linear with Cross Entropy Loss** (Section 2.2.2). Shown in **Section** 4.4.

### 4.2 CONSTRUCTION OF SYNTHETIC DATA: THE 'POSITION WEIGHTED MULTITASK SPARSE PARITY' DATASET

In previous work, a common practice is to mask the leftmost tokens and leave $l$ tokens before the token-to-predict visible to Language Models, as shown in Figure 5. Although this may not show the impact of important tokens to final answer perplexity (e.g., it fails to show the importance of the second key info in Figure 5), this method aligns well with our setting of increasing context length.

Although the next token to predict might depend on several pieces of key information, we see from Figure 5 that the first key token would raise model perplexity.

Inspired by this concept in Figure 5 and the 'multitask sparse parity dataset previously studied in (Michaud et al., 2024; Barak et al., 2022), we propose the 'position-weighted multitask sparse parity dataset. In detail, each input consists of $L$ 'context bits, each bit lies in $\{0, 1\}$. Each subtask takes xor on two certain bits in the context bits, and the answer to some sample is the answer of the only activated subtask, as shown in Figure 5. We use 60 context bits and 200 tasks. From 11th to

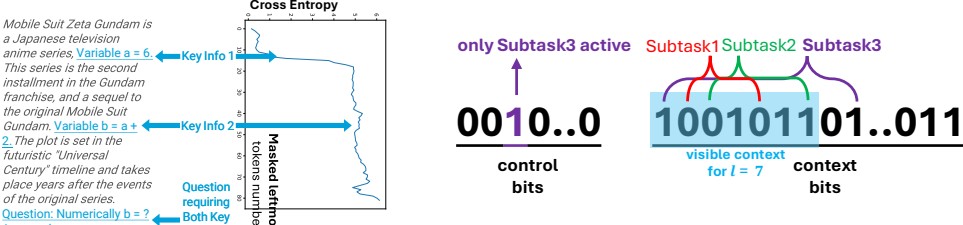

Figure 5: **Left**: An example of the 'two needles in a haystack' task, similar to those in (Levy et al., 2024). The text part is the input to the Language Model, with key information and question visualized in blue; the figure part shows perplexity of the answer token $\langle 8 \rangle$ of LLaMa-3.1-8B (horizontal) vs. number of masked leftmost tokens (vertical). Although seeing both pieces of information are necessary to answer the question, perplexity rises dramatically only when the first piece of information is masked. **Right**: An example of our synthetic data. Each sub-task corresponds to 2 context bits of fixed position. At each time, exactly one sub-task is activate, and the ground truth output is calculated by taking XOR over the 2 context bits of the activate task. As shown in the example, the answer for Subtask 1,2,3 is $0 \oplus 0 = 0, 0 \oplus 1 = 1$ and $1 \oplus 1 = 0$ respectively, but since the thrid bit is 1 for control bits, only Subtask 3 is activated and the final answer is 0. However, for a model of context length 7, it cannot see the $9th$ bit required by subtask 3, making it unable to predict the answer correctly.

the 60th bit, each bit corresponds to the max bit of two tasks: $\#Task|_{max(bit_1,bit_2)=i} = 2, \forall i \in \{11, 12, \ldots, 60\}$.

We assign different frequencies to different tasks, approximating the real-world situation where tasks requiring nearer bits are more often. In all, Bayes Risk, or the minimum Cross Entropy Loss, is:

$$
\begin{aligned}
&R_{Bayes}(ctl) \\
=&MinCELoss(ctl) \\
=&(\sum_{task\ s.t.\ max(bit_1,bit_2)>ctl} freq(task) \log 2)/\sum_{task} freq(task) \\
\approx&A + B/(ctl + C)^{\alpha}
\end{aligned}
$$

More details are shown in Table 1.

### 4.3 TRANSFORMER-BASED SYNTHETIC MODEL WITH ENTROPY MEASUREMENTS

We use a 3-layer causal Transformer, with embedding dimension 208 and FFN dimension 832, RoPE embedding with base frequency 4000; input sequence length is always $60+1$, with 60 context tokens (either 0, 1 or ?) and 1 task tokens (chosen from task tokens of vocab size 200).

We use 100 tasks and 60 task bits. From $11th$ to the $60th$ bit, each bit corresponds to the max bit of two tasks: that is, $\#Task|_{max(bit_1,bit_2)=i} = 2, \forall i \in \{11, 12, \ldots, 60\}$.

During training, $50\%$ of the samples are unmasked, while for the other $50\%$ samples, we mask the last $X$ task bits to be 0.5, where $X$ is a random int from $60 - 10$ to $60 - 60$. This ensures our model to be able to handle mask bits, and also ensures it can learn uncommon tasks (relying on context bits that are at the end of the context bits) well. We train the model on large enough dataset so that it approximates the Bayes Model well (please refer to Table 1 in Appendix for more details).

After the model has been trained, we measure its eigen values, as shown in Figure 7. It is shown that: (1) Larger context length contains more information, hence eigen values in Intrinsic Space degrades slowlier (left figure); (2) the model approximates the theoretical Bayes Model well (as the green points in the middle figure is very close to the orange ones) (middle figure); (3) CE Loss follows a very good linear relationship with sum of log eigenvalues of the first $N$ dimensions for $N \geq 70$ in

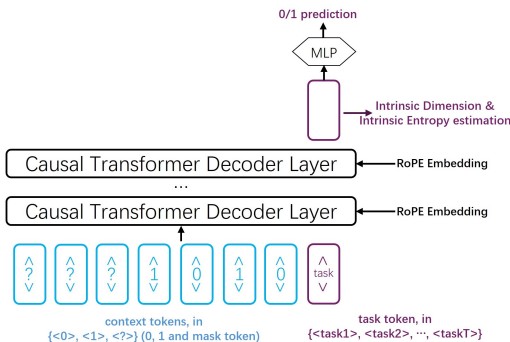

Figure 6: Transformer and Rope-based model for the synthetic task. Here, we use one task token to encode the task information.

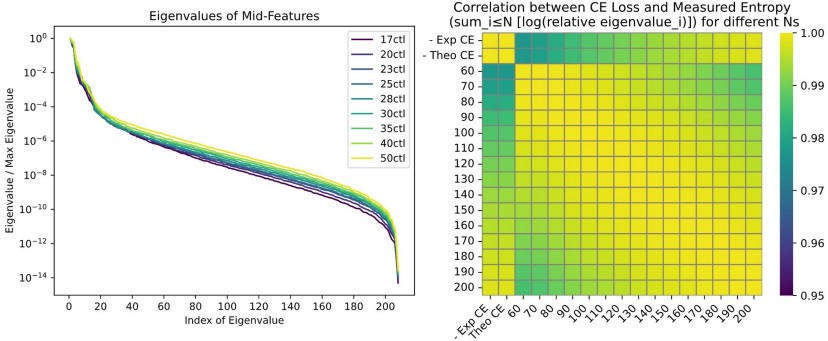

Figure 7: Eigen value and CE results measured on trained model for Synthetic Dataset in this section. **Left**: Eigen Value vs. Index of Eigen Value; **Right**: Correlation between Cross Entropy Loss and Measured Entropy. We see good linear relationship between CE Losses and Measured Intrinsic Entropy from Lower figures.

the Intrinsic Space (right figure), where the case $N = 200$ (all eigen values) are also shown in the middle figure.

This validates **Point 1**: Cross Entropy Loss is approximately linear with Intrinsic Entropy as measured by the sum of log eigenvalues.

## 4.4  ENTROPY IN INTRINSIC SPACE: SYNTHETIC DATASET VALIDATION

Figure 7 shows the measured results of Intrinsic Entropy on the synthetic dataset, which follows a linear relationship with the Cross Entropy Calculated (Theo CE) and Cross Entropy loss measured (Exp CE).

This provides evidence for **Point 2** in Section 4.1: we can measure entropy in the intrinsic space using eigenvalue-based methods or density-based methods, and both show linear relationships with Cross Entropy Loss, validating our entropy-based theoretical framework.

## 5  CONCLUSION AND DISCUSSIONS

### 5.1  CONCLUSION

In this work, we study the impact of context length on language modeling, especially through Bayes Risk and Approximation Loss, from both theoretical and experimental perspectives.

In Section 2, we propose assumptions on the relationship among **CE Loss**, **Intrinsic Entropy**, and **context length**. We derive a linear relation between CE loss and Intrinsic Entropy, and analyze

how context length affects Intrinsic Entropy. We further investigate the relationship among Intrinsic Entropy, context length, and Intrinsic Dimension in **Appendix K** from **an Intrinsic Dimension perspective**. We also provide **formal definitions of assumptions and derivations of key theorems** in **Appendix D**.

We also conduct experiments on both real data (Section 2, Section 3) and synthetic data (Section 4), measuring Intrinsic Entropy and validating the relationship among Cross Entropy loss (Bayes Risk + Approximation Loss), context length, and Intrinsic Entropy.

As a corollary of our theory, an optimal context length exists and increases with dataset size in pretraining; this is validated in Section 3. For downstream tasks such as document QA over long documents, we also observe an optimal context length that increases with task context-length requirements for a fixed model, as shown in Section 3 and Appendix A. We hope our work provides useful insight for future work on long-context language models and on the broader physics of language models.

## 5.2 LIMITATIONS AND FUTURE WORK

Our theory starting from Intrinsic Entropy only holds with assumptions in Section 2; and in Appendix K we use the perspective of Intrinsic Dimension to (partially) explain our assumptions and measurements w.r.t. Intrinsic Entropy. We hope future work may try to propose even more fundamental theories to explain our Intrinsic Entropy measurements.

In our work, similar to several previous work (Bahri et al., 2024; Aghajanyan et al., 2021), we explain the impact of context length scaling from the perspective of Intrinsic Space (or Data Manifold), which is related not only to data, but also potentially to the neural network (that maps the data into such Intrinsic Space) and the prediction task (Bahri et al., 2024). Our explanation leans toward how the model represents the data in its intrinsic space and is hence more related to real language models, meanwhile other types of more model-agnostic explanations might also be proposed.

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

# A  DOWNSTREAM TASK

In the main paper, we experimentally discover and theoretically analyze how context length impact Cross Entropy loss for next token prediction. Previous studies (Hsieh et al., 2024) show that Cross Entropy Loss might not be highly correlated with important downstream tasks.

In this section, we study the impact of context length on downstream document QA tasks, similar to those proposed in Ruler-QA1 (Hsieh et al., 2024). The conclusions we observe in this section are:

- 1. For downstream tasks studied (i.e. similar to Ruler-QA document QA tasks), optimal context length still exists, and this phenomenon can be analyzed from the perspective of Bayes Risk and Approximation Loss.
- 2. For these tasks, Intrinsic Entropy can still act as a proxy of information learned, and when Language Model is not deviating from Bayes Model by a large margin, is still positively-correlated with QA accuracy.

## A.1  OPTIMAL CONTEXT LENGTH ON RULER-QA: A CASE STUDY

Ruler-QA1 (Hsieh et al., 2024) is representative among a series of doc-QA tasks in the sense that (1) it is composed of real-world documents and QA pairs from Doc-QA tasks like SQuAD (Rajpurkar et al., 2016); (2) its samples are generated by inserting a 'golden paragraph' (i.e. the paragraph containing answer to a specific question in SQuAD) into other paragraphs sampled also from SQuAD, hence one can control the total length of tested samples. This provides us with a great testbed for experimenting the impact of visible context length to models, and to test the Intrinsc Entropy. The task shown in Figure 3 (with $\gamma = 0$ setting) aligns with the Ruler-QA1 dataset.

The measured results are shown in Figure 4 in the main paper. We study the performance of Llama-3.1-8B. For the original Ruler-QA1 dataset, the 'golden paragraph' is inserted randomly and uniformly across the sample. We first generate samples with $max\_ctl = 16k$, i.e. each sample has length close to 16k tokens and is composed of multiple paragraphs sampled from SQuAD, and we test the Language Model to answer a question related to some certain paragraph uniformly distributed across the context. During test, we only allow the Language Model to see the closest 'ctl' tokens (and hence it cannot see previous paragraphs), and measure accuracy varying this 'ctl'. The result is shown in the line labeled as 'uniform distributed' in Figure 4. As shown, though the accuracy first increases when increasing context length, but the context length drops after $ctl = 6k$. This proves the existence of an optimal context length, for Llama-3.1-8B on Ruler-QA1.

To further study how this optimal context length depends on the property of tasks, we propose the Position-Weighted Ruler-QA1 dataset. We test the Language Model to answer a question related to a certain paragraph sampled by: $P(x) \propto (1 - x/L)^{\gamma}$, where $x$ is the distance of the paragraph to end of input (in tokens), $L$ is the maximum context length (i.e. $8k$), and $\gamma$ is a hyper-parameter, fixed for certain task. $\gamma = 0$ degrades to uniform distribution (i.e. the standard Ruler-QA1 task), while a larger $\gamma$ means the task focuses less on far-away tokens. Similarly, for a fixed $\gamma$, we adjust the number of tokens visible to Language Models ('ctl') and measure its accuracy; results are also shown in Figure 4. From the figure, we have two observations: (1) an optimal context length exists for each $\gamma$; and (2) a smaller $\gamma$ (i.e. task requires more long context) typically leads to a larger optimal context length.

More detailed analysis of this result from the Bayes Risk and Approximation Loss decomposition perspective can be found in the main paper (Section 3).

## A.2  INTRINSIC ENTROPY: A PROXY OF INFORMATION LEARNED BY LANGUAGE MODEL

We measure Intrinsic Entropy of different context length on the Doc paragraphs samples we construct. In our experiment, we take the closest $ctl$ tokens of the concatenated samples as input to Language Model (Llama-3.1-8B), and take the hidden state of the final layer of a close-to-the-end token. After obtaining $N$ such vectors, we conduct Gaussian-KDE to measure the Intrinsic Entropy. The result is shown in the right figure of Figure 4.

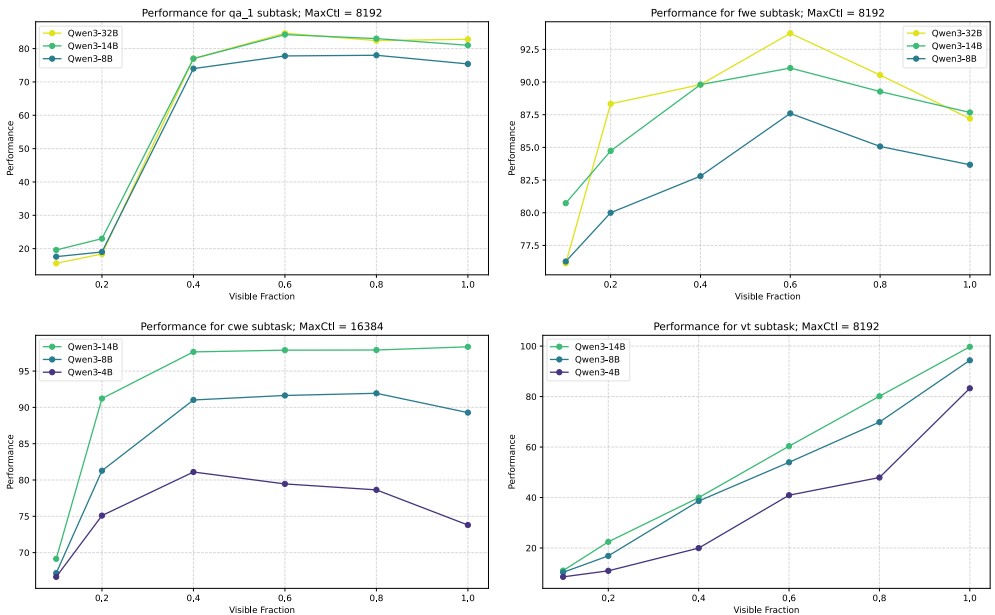

Figure 8: Acc vs. Visible Context Length of Qwen-3 series models (non-thinking chat models) on 4 representative subsets of the RULER dataset: qa_1 (**document qa, upper-left**), fwe (**frequent word extraction, upper-right**), cwe (**common words extraction, lower-left**), and vt (**variable tracking, lower-right**), for a fixed max context length and a varying visible fraction of the input context. Most models show an optimal context length for qa_1, fwe and cwe subtask, while the vt subtask shows increased performance with respect to context length. Moreover, larger model tends to perform better and have a larger optimal context length, represented by the performance comparison between Qwen3-4B and Qwen3-8B on cwe subtask (lower-left).

We observe from Figure 4 that, the Intrinsic Entropy still increases as the input context length increases. Moreover, though measured Intrinsic Entropy does not always follow a linear relationship with QA accuracy (notice that the Intrinsic Entropy calculated does **not** depends on $\gamma$, while QA accuracy is related to the task setting $\gamma$.), we still see a positive correlation between Intrinsic Entropy and QA accuracy when context length is not very long.

In principle, a drop in accuracy when increasing context length actually implies that the model is no longer a good approximation of Bayes Model for certain task at that context length. For relatively larger $\gamma$ tasks (i.e. tasks focusing more on nearer tokens), we see a more aligned trend in increment of QA accuracy and Intrinsic Entropy; while for smaller $\gamma$ tasks (those focusing on farther tokens), the QA accuracy might increase a lot when Intrinsic Entropy increases a little. This potentially implies that Language Models are memorizing and keeping only the information likely to be useful from farther-away tokens, and these pieces of information are sufficient for the QA task.

## A.3 MORE EXPERIMENTS ON RULER BENCHMARK

To see how different tasks might have different behaviors with respect to context length, we further conduct experiments on three RULER subtasks: the qa1 subtask (document qa), the cwe subtask (i.e. common words extraction), the vt subtask (i.e. variable tracking), and the fwe subtask (i.e. frequent words extraction). Other subtasks like single-needle-in-haystack are too simple for sota LLMs hence we did not perform experiments on them.

To study the impact of model size, we utilize the Qwen3 series models. We use the non-thinking mode of the chat models of Qwen3 series Yang et al. (2025), including Qwen3-4B, Qwen3-8B, QWen3-14B and Qwen3-32B for experiments. We use codebase modified from RULER Hsieh et al. (2024). The maximum context length is set to $16k$ for cwe and $8k$ for other subtasks. These results are shown in Figure 8.

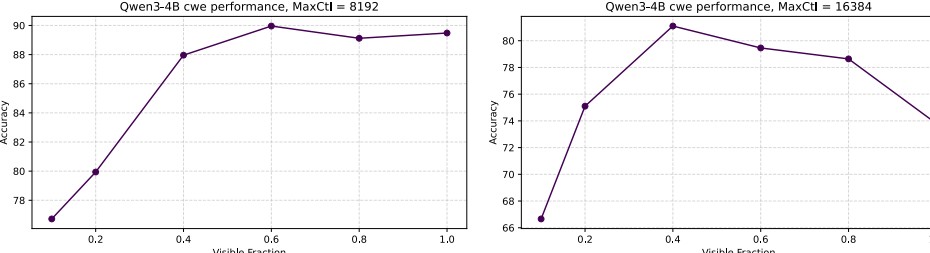

Figure A.3-2. Acc vs. Visible Context Length of Qwen-3 4B, on cwe (common words extraction) subtask, with different Max context length **(left:** $MaxCtl = 8k$**, right:** $MaxCtl = 16k$**)**. As shown, though optimal context length is hard to observe for the task requiring $8k$ as max context length, it is easy to observe for that requiring $16k$ as max context length.

As shown in Figure 8, **(1)** for subtasks resembling fwe, document qa, variable tracing, etc., there exists an optimal context length for most models tested; and **(2)** for vt (variable tracking), in the experiment we conducted models' performance improve with respect to visible context fraction. This could be caused by the fact that variable tracking is relatively easy for current LLMs, thus their approximation loss is low; while given its distribution of variables the Bayes Risk would constantly decrease with more context length, thus it shows a trend of improving even for long visible fraction.

Comparing the results of Qwen3-4B, Qwen3-8B and Qwen3-14B on the cwe subtask in Figure 8, we see that: **(1) optimal context length is larger and harder to observe for larger models**, this can be attributed to larger models lead to less Approximation Loss; comparing results of the same cwe subtask on different max context lengths in Figure A.3-2, we see that: **(2) optimal context length is easier to observe for longer task**, which can also be attributed to a larger Approximation Loss (i.e. language models fall short to deal with longer contexts).

These results could potentially argue that the subtasks defined in RULER are of different difficulty for current LLMs. That is, if one can observe an optimal context length, this proves that the model **gets distracted** by more context beyond the optimal context length, and hence performance gets worse even if these context contains more information. **Potentially, our work provides a new perspective:** Consider a case where some specific LM achieves $95\%$ accuracy on certain subtask with $0.8$ visible context fraction and $90\%$ accuracy on it with $1.0$ visible context fraction. Even though the absolute accuracy numbers are high, the existence of optimal context length and degraded performance also implies an ineffectiveness of the language model when handling long contexts with respect to that specific task.

# B   DEFINITION AND PROPERTIES OF CROSS ENTROPY LOSS

## B.1   DEFINITION OF CROSS ENTROPY LOSS DISCUSSED IN THIS WORK

It is well-known that the original definition of Cross Entropy between two sequential distributions $P$ and $Q$: $H_{org}(P, Q)$ should be:

$$H_{org}(P, Q) = \sum_x - P(x) \log Q(x)$$
$$= \sum_x - P(x_{-\infty:0}) P(x_0 | x_{-\infty:0})$$
$$* \log\{Q(x_0 | x_{-\infty:0}) Q(x_{-\infty:0})\},$$

where $x_{a:b}$ denotes $x_a, x_{a+1}, \ldots, x_{b-1}$; it is common practice to calculate perplexity of Language Models with its input as GT labels (e.g. in technical report of LLaMa-3(Grattafiori et al., 2024)), in other words, the experimentally measured Cross Entropy $H_{exp}(P, Q)$ is actually:

$$H_{exp}(P,Q)$$
$$=\sum_x -P(x_{-\infty:1})\log\{Q(x_0|x_{-\infty:0})\mathbf{P}(\mathbf{x}_{-\infty:0})\}$$
$$=\text{Const} + E_{x_{-\infty:0}}\sum_{x_0} -P(x_0|x_{-\infty:0})\log Q(x_0|x_{-\infty:0}).$$

Therefore, in this work we use:

$$H(P,Q)$$
$$=H_{exp}(P,Q)$$
$$=\mathrm{E}_{x_{-\infty:0}}[\sum_{x_0} -P(x_0|x_{-\infty:0})\log Q(x_0|x_{-\infty:0})] \tag{6}$$

as the definition of Cross-Entropy loss, and $P(x_0|x_{-\infty:0})$, $Q(x_0|x_{-\infty:0})$ as the definition of Natural Language distribution and Language Model distribution, respectively.

### B.2 CROSS ENTROPY LOSS FOR LANGUAGE MODEL WITH CONTEXT LENGTH $l$

In Equation B.1, if $Q_l(x_0|x_{-\infty:0})$ is a language model with limited context length $l$: $Q_l(x_0|x_{-\infty:0}) = Q_l(x_0|x_{-l:0})$, we have:

$$H(P,Q_l)$$
$$=\mathrm{E}_{x_{-\infty:0}}[\sum_{x_0} -P(x_0|x_{-\infty:0})\log Q_l(x_0|x_{-l:0})]$$
$$=-\sum_{x_{-\infty:1}} P(x_{-\infty:1})\log Q_l(x_0|x_{-l:0})$$
$$=-\sum_{x_{-\infty:-l}}\sum_{x_{-l:1}} P(x_{-\infty:-l}, x_{-l:1})\log Q_l(x_0|x_{-l:0})$$
$$=-\sum_{x_{-l:1}} P(x_{-l:1})\log Q_l(x_0|x_{-l:0})$$
$$=\mathrm{E}_{x_{-l:0}}[\sum_{x_0} -P(x_0|x_{-l:0})\log Q_l(x_0|x_{-l:0})]$$
$$=H(P_l,Q_l).$$

Note that $P(x_0|x_{-l:0})$ is exactly the Bayes Model with context length $l$. Hence, we have:

$$D_{KL}(P,Q_l) = -H(P) + H(P,Q_l)$$
$$= -H(P) + H(P_l,Q_l)$$
$$= -H(P) + H(P_l) + D_{KL}(P_l,Q_l).$$

Specially, if we are calculating the KL Divergence between Nature Language and Bayes Model with context length $l$, thus $Q_l = P_l$, we have:

$$D_{KL}(P,P_l) = -H(P) + H(P_l,P_l) = -H(P) + H(P_l). \tag{7}$$

## C EXPERIMENTALLY MEASURE NEXT-TOKEN-PREDICTION INFORMATION ENTROPY $S_{ntp}$

### C.1 PCA-BASED INFORMATION ENTROPY ESTIMATION

Though related, Entropy in Intrinsic Space does not equal to Entropy in the next token prediction task. From the probability perspective, let $dec(x)$ be the next decoded token for some point $x$ in the

intrinsic space, we have: $S = \sum_{x \in IS} -P(x) \log P(x)$, while $S_{ntp} = -\sum_{v \in vocab} P(v) \log P(v)$ where $P(v) = \sum_{x \in IS, dec(x)=v} P(x)$. $S_{ntp}$ is a coarse-grained Entropy compared to $S$. $S$ contains important information on previous tokens that are important for the prediction of future tokens, while $S_{ntp}$ is related only to the next token.

Experiments in Figure 14 show that, no matter what subspace we use, the Cross Entropy Loss usually follows a linear relationship with the Entropy we measured in the subspace, **supporting the claim that the next token prediction task likely lies in some subspace of the Intrinsic Space, or (statistically) its Entropy should be some weighted average of Entropy of several subspaces of similar dimension.** This also suggests that $H_{ntp}$ is approximately linear with $H_{IS}$, which validates our previous assumptions and claims.

## C.2 GAUSSIAN-KDE BASED INFORMATION ENTROPY ESTIMATION

In this sub-subsection we use another method for Information Entropy Estimation. As shown in Figure 9, this estimation also aligns well with PCA-based estimation; moreover, such estimated entropy is also linear with respect to Intrinsic Dimension and Cross Entropy Loss.

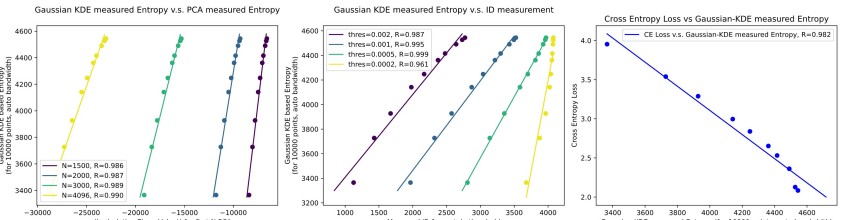

Figure 9: Gaussian-KDE measured Entropy (10000 samples, auto bandwidth = 0.997756) vs. PCA-measured Entropy (left), Measrued ID (middle) and Cross Entropy Loss (right).

## C.3 SYNTHETIC DATASET: ENTROPY IN INTRINSIC SPACE, AND ENTROPY FOR OUTPUT LAYER

For our synthetic dataset, if we view the **Context Feature Vector** shown in **Figure 17** as the feature in the Intrinsic Space, then the best strategy for the context encoder is to generate the answer for all subtasks (it can see) in the Intrinsic Space (since it cannot see the task bits). This would lead to an Entropy of $S = T \log 2$ in the Intrinsic Space.

The entropy of the output layer is, however, $S_{output} = \log 2$ since the answer bits $0, 1$ have the same probability. In this way, the answer of the output layer actually corresponds to one dimension in the Intrinsic Space, which should be the exact dimension at which the answer of the current task is stored. Therefore, $S_{output} = 1/T * S_{IS}$, which explains why the Entropy for output logits is linear to the Entropy for Intrinsic Space.

## C.4 DETAILS FOR SYNTHETIC DATA

Here we present details for synthetic dataset and model training.

We train the model on a training set of 10000000 and a validation set of size 1000000, for 125 epochs (and an early stopping setting of 25 epochs, though the training process did not trigger early stopping).

To make sure that the trained model can be used to approximate the Bayes Model, we compare the model's loss on validation set with context $ctl$ with the calculated minimum possible CE Loss for the task. As shown in Table 1 that the model is not too different from the Bayes Model: the BCE Loss only differs by around 0.001. **Thus, we can use the middle-representation (shown as context feature in Figure 6) as the feature in Intrinsic Space to approximate the Bayes Model for** $17 \le ctl \le 50$**.**

$$MinCELoss(ctl) = (\sum_{\text{task s.t. } max(bit_1, bit_2) > ctl} freq(task) * \log 2)/\sum_{task} freq(task)$$

| Context Length | Model CE Loss | Minimum CE Loss Calculated |
|:---:|:---:|:---:|
| 17 | 0.4648 | 0.4643 |
| 20 | 0.3988 | 0.3988 |
| 23 | 0.3438 | 0.3437 |
| 25 | 0.3119 | 0.3116 |
| 28 | 0.2687 | 0.2686 |
| 30 | 0.2429 | 0.2429 |
| 35 | 0.1867 | 0.1864 |
| 40 | 0.1390 | 0.1387 |
| 50 | 0.0613 | 0.0612 |

Table 1: Comparison between trained model and Bayes Model (minimum CE Loss) for Synthetic Data

## D  DEFINITIONS OF INTRINSIC SPACE AND DERIVED PROPERTIES

As mentioned in Section 2.1.2, in previous work (Bahri et al., 2024; Cheng et al., 2023), as a common practice, the 'Data Manifold' is often **defined** as the middle feature representation of well-trained neural networks, and **assumptions** are made on this kind of mid-representation, with experiments to **validate** these assumptions. (Intrinsic Space is defined as the space where the Data Manifold lies.)

Meanwhile, the Data Manifold can be more formally **defined** by a mapping from input data to some Intrinsic Space which satisfies a certain set of **properties**, and mid-representation of well-trained neural networks are **assumed to have such properties**, which can be experimentally **validated**. These two perspectives are actually, equivalent to each other:

- **Perspective 1**: Experiments show mid-representations of neural networks have certain properties → Data Manifold in Intrinsic Space satisfies such properties.
- **Perspective 2**: Experiments show mid-representations of neural networks have certain properties → such mid-representation can be viewed as Data Manifold of Intrinsic Space that is defined to have such properties.

These two perspectives are equivalent to each other, and Perspective 1 is used in some previous work (Bahri et al., 2024; Sharma & Kaplan, 2022).

In this section, we formally define the Intrinsic Space and formally derive related results, following **Perspective 2**.

### D.1  FORMAL DEFINITIONS OF INTRINSIC SPACE

We define an *intrinsic space* to formalize the latent structure underlying natural language sequences. This space is independent of surface forms and aims to capture the semantic and syntactic essence of language contexts across different sequence lengths.

**Setup.**  Let $\mathcal{V}$ be a finite vocabulary and $\mathcal{X} = \mathcal{V}^*$ the set of all finite sequences over $\mathcal{V}$. Let $\mathcal{M} \subset \mathcal{X}$ denote the *original data manifold* of natural language, i.e., the support of the data distribution $p(x)$.

**Definition.**  An **intrinsic space** $\mathcal{Z}$ is a latent representation space defined by a mapping

$$\Phi : \mathcal{X}_{\leq t} \to \mathcal{Z}, \tag{8}$$

where $\mathcal{X}_{\leq t} = \bigcup_{k=0}^{t} \mathcal{V}^k$ is the set of all language contexts of length $t$. The image of the *original data manifold* under this map is denoted $\mathcal{M}_{\mathcal{Z}} = \Phi(\mathcal{M}_{\leq t}) \subset \mathcal{Z}$, or the *data manifold* (in Intrinsic Space). We require the following properties:

- **Predictive Consistency:** There exists a decoder $\pi : \mathcal{Z} \to \Delta(\mathcal{V})$ such that

$$\pi(\Phi(x_{<t})) = p(x_t \mid x_{<t}), \tag{9}$$

  i.e., the intrinsic representation enables accurate next-token prediction.

Moreover, there are some other properties assumed (separately) in our work, for which we give a formal definition here.

- **Uniform Information Gain:** This assumption assumes the following linear relationship between predictive divergence and intrinsic dimension $dim(l)$:

$$D_{KL}(P, P_l) = s \cdot (\dim(\infty) - \dim(l)) \tag{10}$$

  for some constant $s > 0$, which we interpret as the average number of bits of predictive information contributed by each intrinsic dimension. This is empirically observed in experiments.

- **Linear Entropy Relationship** This assumption assumes that there exists constant $0 < s < 1, b$ and a sequence of tolerances $\{\varepsilon_t\}_{t \geq 0}$ with $\varepsilon_t \to 0$ such that for every context length $t$,

$$\left| -s\, H\big[q_t(Z)\big] + b - H\big[p(\cdot \mid x_{<t})\big] \right| \leq \varepsilon_t. \tag{11}$$

  Where $q_t(\cdot)$ denotes probability density function. Equivalently, in the idealized zero-tolerance limit:

$$-s\, H\big[q_t(Z)\big] + b = H\big[p(x_t \mid x_{<t})\big] \quad \forall\, t. \tag{12}$$

  It is worth mentioning that, we can easily derive the linear entropy relationship from the uniform information gain assumption, but not vice versa. Hence, linear entropy relationship is a *weaker* assumption compared to uniform information gain.

- **Lipschitz Differentiable Density** This assumption assumes the density of data distribution is smooth in the intrinsic space:

$$\|\nabla q(z)\| \leq L \tag{13}$$

  for some constant $L > 0$

- **Finite $\epsilon$-negative moment:** This assumption means the integral of the $\epsilon$-negative moment of the data distribution is finite:

$$\int_{\mathcal{Z}} q(z)^{1-\epsilon}\, dz := C_\epsilon < \infty. \tag{14}$$

  Remark. When $\mathcal{Z}$ is bounded ($\int_{\mathcal{Z}} q(z)\, dz = V_{\mathcal{Z}} < \infty$), and if there exists a constant $q_{min} > 0$ s.t. $q(z) \geq q_{min} > 0$, then this assumption is satisfied. Hence this is a weaker assumption compared to boundedness and non-zero density, which is even weaker than uniform distribution assumption.

To conclude: if some space satisfies these properties, then the data representation is referred to as 'Data Manifold' in this space, and such properties would lead to further derivations in these work (including this work). In experiments, Middle-representation of neural networks are assumed (and shown) to have these kind of properties, hence explain some of the scaling behaviors they have.

## D.2 DERIVATION FOR DATA SCALING FOR APPROXIMATION LOSS

**Theorem 1** (Expected capped nearest–neighbour distance). *Let $\mathcal{Z} \subseteq \mathbb{R}^d$ ($d \geq 1$) and there exists a non-empty open set $U \in \mathbb{R}^d$ such that $U \subseteq \mathcal{Z}$ (i.e., $\mathcal{Z}$ is a $d$-dimensional region). Let $q : \mathcal{Z} \to [0, \infty)$ be a probability density satisfying*

1. ***Lipschitz Differentiable:** $\|\nabla q(z)\| \leq L$ for all $z \in \mathcal{Z}$;*

2. ***Finite $\epsilon$-negative moment:** for some fixed $\epsilon > 1/d$, $\int_{\mathcal{Z}} q(z)^{1-\epsilon}\, dz := C_\epsilon < \infty.$*

| | Key Properties Assumed | Derived Results |
|---|---|---|
| (Bahri et al., 2024)Theorem 2 | Bounded, Uniform Distribution, Lipschitz Differentiable | Data Scaling for Approx. Loss |
| Theorem 1, 2 in Appendix D.2 (corr. to Section 2.3) | (Bounded,) Finite Negative Moment, Lipschitz Differentiable | Data Scaling for Approx. Loss |
| Theorem 3 in Appendix D.3 (corr. to Section 2.2.1) | Predictive Consistency, Uniform Information Gain | Bayes Risk for Ntp of varied Ctl (Intrinsic Dimension perspective) |
| Theorem 4 in Appendix D.4 (corr. to Section 2.2.1) | Predictive Consistency, Linear Entropy Relationship | Bayes Risk for Ntp of varied Ctl (Information Entropy perspective) |

Table 2: **In this Section (Appendix D) we formulate results from previous sections with Theorems derived with defined assumptions and properties of intrinsic space in this section.** Ntp refers to Next-token-prediction, Ctl refers to Context Length. We derive data scaling for approximation loss with weaker assumptions compared to (Bahri et al., 2024), please refer to Theorem 1, 2 in Appendix D.2 for more details.

*Draw i.i.d. samples $\mathcal{Z}_D = \{Z_1, \ldots, Z_D\} \sim q^{\otimes D}$ and define the capped nearest–neighbour distance*

$$R_M(Z_i) = \min\{ M, \min_{j \neq i} \|Z_i - Z_j\| \}, \qquad M > 0. \tag{15}$$

*Then, there exists constant $C = C(d, L, \epsilon, M, C_\epsilon)$ and $D_0 = D_0(d, L, \epsilon, M, C_\epsilon)$ such that $\forall D > D_0$:*

$$\mathbb{E}_{\mathcal{Z}_D}\big[R_M(Z_1)\big] \leq \begin{cases} C (\log D)^{\epsilon/(d+1)} D^{-\epsilon/(d+1)}, & \text{if } \epsilon < \dfrac{d+1}{d}, \\ C D^{-1/d}, & \text{if } \epsilon \geq \dfrac{d+1}{d}. \end{cases} \tag{16}$$

*Thus, there exists constant $c = c(L, \epsilon, M, C_\epsilon)$, such that:*

$$\mathbb{E}_{\mathcal{Z}_D}\big[R_M(Z_1)\big] \leq C D^{-c/d}. \tag{17}$$

*Proof.* We write $Z_1$ for the distinguished point and $R(Z_1) = \min_{j \neq 1} \|Z_1 - Z_j\|$ for its exact nearest–neighbour distance, always capping by $M$ at the very end. Throughout the proof the expectation $\mathbb{E}[\cdot]$ is taken over the whole sample $\mathcal{Z}_D = (Z_1, \ldots, Z_D) \sim q^{\otimes D}$.

**Step 0. Notation.**

$$v_d := \text{vol}(B_1(0)), \qquad c_d := \frac{v_d}{2}, \qquad r_0(z) := \frac{q(z)}{2L}.$$

$v_d$ is the volume of a unit ball. Moreover, since $\|\nabla q\| \leq L$, whenever $r \leq r_0(z)$ one has $q(u) \geq \frac{1}{2}q(z)$ for every $u \in B_r(z)$.

**Step 1. Exponential hole probability inside the Lipschitz ball.**

Fix $z$ and $r \leq r_0(z)$. The *mass* of $q$ inside $B_r(z)$ satisfies

$$\mu_r(z) := \int_{B_r(z)} q(u)\,du \geq \frac{1}{2}q(z)\,v_d r^d = c_d q(z) r^d.$$

Conditioned on $Z_1 = z$, the $(D-1)$ other points are i.i.d. $q$, so the conditional probability that all other points are sampled outside the ball $B_r(z)$ is:

$$\Pr\big(R(z) > r \,\big|\, Z_1 = z\big) = \big(1 - \mu_r(z)\big)^{D-1} \leq \exp\big[-c_d(D-1)q(z)\,r^d\big]. \tag{18}$$

**Step 2. Density threshold and spatial split.**

Define a data–dependent threshold

$$\lambda_D := \Big(\frac{2^{d+1}L^d d \log D}{c_d D}\Big)^{\frac{1}{d+1}} \quad (D \geq 2), \qquad \mathcal{H}_D := \big\{z : q(z) \geq \lambda_D\big\}, \quad \mathcal{L}_D := \mathbb{R}^d \backslash \mathcal{H}_D.$$

(The power $1/(d+1)$ is tuned to balance two error terms below.)

### Step 3.1. Distribution in $\mathcal{H}_D$ (moderate or high density region).

For every $z \in \mathcal{H}_D$ set

$$\rho(z, D) := \Big(\frac{2d \log D}{c_d D \, q(z)}\Big)^{1/d} .$$

*Bound on $\rho$.* Since $q(z) \geq \lambda_D$, $\rho(z, D) \leq r_0(z)$ and therefore equation 18 is valid for all $0 < r \leq \rho(z, D)$.

*Tail probability at $\rho$.* With $r = \rho(z, D)$,

$$\Pr\big(R(z) > \rho(z, D) \,\big|\, Z_1 = z\big) \leq \exp[-2d \log D] = D^{-2d}.$$

Because $R_M(Z_1) = \min(R(Z_1), M) \leq M$,

$$\mathbb{E}\big[R_M(Z_1)\mathbf{1}_{\{R > \rho\}} \,\big|\, Z_1 = z\big] \leq M D^{-2d}. \tag{19}$$

*Integral of $R$ up to $\rho$.* Using equation 18,

$$\mathbb{E}\big[R(Z_1) \wedge \rho(z, D) \,\big|\, Z_1 = z\big] = \int_0^\rho \Pr(R > r \,|\, Z_1 = z)\, dr$$

$$\leq \int_0^\rho \exp\big[-c_d(D-1)q(z)r^d\big]\, dr.$$

Make the change of variable $t := c_d(D-1)q(z)r^d$; then $r = (t/c_d(D-1)q(z))^{1/d}$ and $dr = \frac{1}{d}\, r\, dt/t$. The upper limit $r = \rho$ maps to $t = 2d \log D$. Hence

$$\int_0^\rho \exp[-c_d(D-1)q(z)r^d]\, dr = \frac{\Gamma(1+1/d)}{d^{1/d}c_d^{1/d}}\big(Dq(z)\big)^{-1/d}.$$

Absorbing constants:

$$\mathbb{E}\big[R(Z_1) \wedge \rho(z, D) \,\big|\, Z_1 = z\big] \leq C_{d,L}\big(Dq(z)\big)^{-1/d}. \tag{20}$$

*Average over $z \in \mathcal{H}_D$.* Taking expectation over $Z_1$ first restricted to $\mathcal{H}_D$ and then combining equation 20 with equation 19,

$$\mathbb{E}\big[R_M(Z_1)\mathbf{1}_{\mathcal{H}_D}(Z_1)\big] \leq C_1 D^{-1/d} + M D^{-2d}, \quad C_1 := C_{d,L}\big(\mathbb{E}[q(Z)^{-1/d}]\big)^{1/d} < \infty. \tag{21}$$

### Step 3.2. Distribution in $\mathcal{L}_D$ (ultra–low density region).

On $\mathcal{L}_D$ one has $q(z)^\epsilon \leq \lambda_D^\epsilon$, so by Hölder's inequality

$$\Pr\big(Z \in \mathcal{L}_D\big) = \int_{q < \lambda_D} q(z)\, dz \leq \lambda_D^\epsilon \int_{\mathbb{R}^d} q(z)^{1-\epsilon}\, dz = C_\epsilon \lambda_D^\epsilon.$$

Since $R_M \leq M$,

$$\mathbb{E}\big[R_M(Z_1)\mathbf{1}_{\mathcal{L}_D}(Z_1)\big] \leq M C_\epsilon \lambda_D^\epsilon = M C_\epsilon\big(\log D\big)^{\epsilon/(d+1)} D^{-\epsilon/(d+1)}. \tag{22}$$

### Step 4. Global bound.

Adding equation 21 and equation 22. For large enough $D$, the term $MD^{-2d}$ is higher-order small quantity compared to $D^{-1/d}$ or $D^{-\epsilon/(d+1)}$. Therefore,

$$\mathbb{E}\big[R_M(Z_1)\big] \leq C_1 D^{-1/d} + M C_\epsilon(\log D)^{\epsilon/(d+1)} D^{-\epsilon/(d+1)} + o(\min(D^{-1/d}, D^{-\epsilon/(d+1)})).$$

Finally, compare the two powers of $D$. If $\epsilon < \frac{d+1}{d}$ then $\epsilon/(d+1) < 1/d$ and the second term dominates; otherwise the first dominates. This yields the two–case estimate claimed. $\qquad \square$

**How large can $\epsilon$ be, for unbounded and bounded $\mathcal{Z}$?**

- **It is usually assumed (Bahri et al., 2024; Sharma & Kaplan, 2022) that $\mathcal{Z}$ is bounded**: this assumption makes sense since in usual cases we approximate Intrinsic Space with middle feature representation of neural networks, which can indeed be bounded. For bounded $\mathcal{Z}$, it is possible for $\epsilon$ to be larger than $1 + 1/d$. We would like to mention that **in (Bahri et al., 2024), a constant distribution $q(z) = Const$ is assumed, where $\epsilon$ can be arbitrarily large and dominant rate $D^{-1/d}$ is derived: this is a much stronger assumption compared to Finite $\epsilon$-negative moment we assumed in our work.**

- **If $\mathcal{Z}$ is bounded and $\exists q_{min} > 0$ such that $\forall z, q(z) \geq q_{min} > 0$, then $\forall z > 1$,** $\int_{\mathcal{Z}} q(z)^{1-\epsilon} dz \leq \int_{\mathcal{Z}} q_{min}^{1-\epsilon} dz = q_{min}^{1-\epsilon} \int_{\mathcal{Z}} dz$, and the final term is finite for any $\epsilon$. That is, in this case $\epsilon$ can be arbitrarily large.

- For unbounded $\mathcal{Z}$, $\epsilon < 1$ is the usual case; at $\epsilon = 1$ the condition becomes $\int q^0 = $ Leb$(\text{supp } q) < \infty$, i.e. *compact support of finite measure*. For most unbounded densities (Gaussians, sub-exponential, power-law) one only has $\epsilon < 1$.

- The comparison threshold $\frac{d+1}{d}$ is always $> 1$ when $d \geq 1$; hence the dominant rate is

$$\begin{cases} D^{-\epsilon/(d+1)} & \text{for every admissible } 1/d < \epsilon < 1, \\ D^{-1/d} & \text{only if the support is compact and } \epsilon > 1 + 1/d. \end{cases} \tag{23}$$

Thus $\epsilon$ can never "reach" the critical value $\frac{d+1}{d}$ unless $q$ is essentially bounded below on its support.

**From Nearest-Neighbour Distance to Approximation Loss**

- Capped nearest-neighbour distance can be derived natually if one assume the maximum distance of neighboring points to be bounded by constant, or if one assume the Intrinsic Space $\mathcal{Z}$ is bounded.

- **Restate of Theorem 2 in (Bahri et al., 2024)**: Assuming $l(f), f, F$ be Lipschitz with contants $K_L, K_f, K_F$ and $l(F) = 0$, $D$ be training dataset of size $D$ sampled i.i.d from $M_d$. Let $f(x) = F(x) \forall x \in D$. Then, for each training point $x$, let $\hat{x}$ be the nearest neighboring training data point, we have $L(D) \leq K_L(K_f + K_F)\mathbb{E}_{D,x}[||x - \hat{x}||]$.

- Combining **Theorem 2 in (Bahri et al., 2024)** and previous results (nearest neighbour distance in this Appendix D.2), since Approximation loss of context length $l$ is $D_{KL}(P_l, Q)$ which can be 0 when $Q = P_l$, thus satisfying the assumption of **Theorem 2 in (Bahri et al., 2024)**. Thus, $L_{Approx} = C_0 + A(l)/D^{c/dim} = C_0 + A(l)/D^{\alpha(l)}$.

Therefore, we have:

**Theorem 2** (Data Scaling for Approximation Loss). *Let $\mathcal{Z} \subseteq \mathbb{R}^d$ $(d > 1)$ be a d-dimensional region (exists non-empty open set $U \in \mathbb{R}^d$ such that $U \subseteq \mathcal{Z}$). $q : \mathcal{Z} \to [0, \infty)$ be probability density function satisfying **Lipschitz Differentiable** and **Finite $\epsilon$-negative moment**. Let $g : \mathcal{Z} \to P_V$ be a decoding mapping from intrinsic space $\mathcal{Z}$ to a distribution of tokens in vocabulary $V$, and $l(P_{V1}, P_{V2})$ be KL divergence loss function (thus $l$ is zero for identical distributions). Assume $l(g(z_1), g(z_2))$ is differentiable and Lipschitz smooth for $z_1$ and $z_2$ with Lipschitz coefficient $L_l$.*

*Then, draw i.i.d. samples $\mathcal{Z}_D = \{Z_1, \ldots, Z_D\} \sim q^{\otimes D}$, if $\min_{j \neq i} ||\mathcal{Z}_i - Z_j||$ is bounded by $M$, then there exists constant $C = C(d, L, \epsilon, M, C_\epsilon, L_l)$ and $c = c(L, \epsilon, M, C_\epsilon)$ such that for $D > D_0(d, L, \epsilon, M, C_\epsilon)$:*

$$\min_{j \neq i} l(g(Z_i), g(Z_j)) \leq C \, D^{-c/d}. \tag{24}$$

*Proof.*

$$\begin{aligned} \min_{j \neq i} l(g(Z_i), g(Z_j)) &\leq \min_{j \neq i} L_l \cdot ||Z_i - Z_j|| \\ &= L_l \cdot \min\{M, \min_{j \neq i} ||Z_i - Z_j||\} \\ &= L_l \cdot R_M(Z_i) \text{ by the definition of } R_M(Z_i) \text{ in Theorem 1} \end{aligned} \tag{25}$$

By applying Theorem 1 we have:

$$\mathbb{E}_{\mathcal{Z}_D}[\min_{j \neq i} l(g(Z_i), g(Z_j))] \leq L_l \, C \, D^{-c/d} \tag{26}$$

for constant $C = C(d, L, \epsilon, M, C_\epsilon)$, $c = c(L, \epsilon, M, C_\epsilon)$ and large enough $D > D_0(d, L, \epsilon, M, C_\epsilon)$, thus completing the proof. $\qquad\square$

**Meaning of a finite $\epsilon$-negative moment**

- **Lebesgue–measure view**

  Write $E_t := \{z : q(z) \leq t\}$. Chebyshev gives

  $$\text{Leb}(E_t) \leq t^{-\epsilon} \int q^{1-\epsilon} = C_\epsilon \, t^{-\epsilon}. \tag{27}$$

  Hence Assumption (A2) controls how *large* the very–low–density region can be; the smaller $\epsilon$, the larger that region may grow.

- **Rényi entropy view**

  For order $\alpha > 0$, the Rényi entropy is

  $$H_\alpha(q) = -\frac{1}{\alpha - 1} \log \int q^\alpha. \tag{28}$$

  Setting $\alpha = 1 - \epsilon \in (0, 1)$ (the *Tsallis* regime) and re-arranging,

  $$\int q^{1-\epsilon} = e^{-(1-\epsilon)H_{1-\epsilon}(q)}, \tag{29}$$

  so finiteness of the $\epsilon$-negative moment is equivalent to *finite sub-Rényi entropy of order* $< 1$. Smaller $\epsilon$ (order closer to 1) corresponds to heavier low-density tails, which precisely slows the nearest-neighbour rate as captured in Theorem 1.

### D.3 DERIVATION FOR BAYES RISK WITH INTRINSIC DIMENSION ASSUMPTION

**Theorem 3** (Bayes Risk and Context Length with Intrinsic Dimension Assumption). *Let $\mathcal{Z}$ be an intrinsic space satisfying **Predictive Consistency** and **Uniform Information Gain**, then the Bayes Risk $H(P, P_l)$ of context length $l$ is **Linear** with respect to Intrinsic Dimension $dim(l)$. That is,*

$$H(P, P_l) = -s \cdot dim(l) + Const \tag{30}$$

*Proof.*

$$\begin{aligned} H(P, P_l) &= H(P) + D_{KL}(P, P_l) \\ &= H(P) + s \cdot (dim(\infty) - dim(l)) \\ &= -s \cdot dim(l) + Const \end{aligned} \tag{31}$$

$\qquad\square$

**An intuitive example for the 's-bits per dimension' assumption**: assuming that the vocab is an integer from 0 to $2^{dim(\infty)*s} - 1$. assuming $P(x_0|x_{-\infty:0}) = \delta_{x_0,y}$, that is, the next token given $x_{-\infty:0}$ is sure to be $y$. $y$. For $P_l(x_0|x_{-\infty:0})$, the first $dim(l) * s$ digits of the integer (in binary representation) are known, but the remaining $(dim(\infty) - dim(l)) * s$ digits are unknown, making a guess in these numbers yield $P_l(x_0|x_{-\infty:0}) = 1/2^{s*(dim(\infty)-dim(l))}$. Thus, $D_{KL,x_0}(P(x_0|x_{-\infty,0}), P_l(x_0|x_{-\infty,0})) = 1 * \log 1/(1/2^{s*(dim(\infty)-dim(l))}) = s * (dim(\infty) - dim(l))$.

### D.4 Derivation for Bayes Risk with Information Entropy Assumption

**Theorem 4** (Bayes Risk and Context Length with Information Entropy Assumption). *Let $\mathcal{Z}$ be an intrinsic space satisfying **Predictive Consistency** and **Linear Entropy Relationship** of zero-tolerance limit, then the Bayes Risk $H(P, P_l)$ of context length $l$ is **Linear** with respect to Intrinsic Entropy $H[q_t(\mathcal{Z})]$ where $q_t(\cdot)$ denotes probability density function. That is,*

$$H(P, P_l) = -s \cdot H[q_t(Z)] + Const \tag{32}$$

*Proof.*

$$\begin{aligned}
H(P, P_l) &= H(P_l) \text{ (from Appendix B.2)} \\
&= H[p(x_t | x < t)] \\
&= -s \cdot H[q_t(Z)] + b \\
&= -s \cdot H[q_t(Z)] + Const
\end{aligned} \tag{33}$$

$\square$

## E More experiments of LLaMa on another dataset

According to the technical report of LLaMa 3.1(Grattafiori et al., 2024), the text corpora with number of 'dirty words' beyond certain threshold would be filtered out, as proposed in (Raffel et al., 2023). We collect some text corpora online which include forbidden words defined in (Raffel et al., 2023), as text corpora unseen by LLaMa 3.1. By conducting experiments on it we obtain results similar to Openwebtext subset.

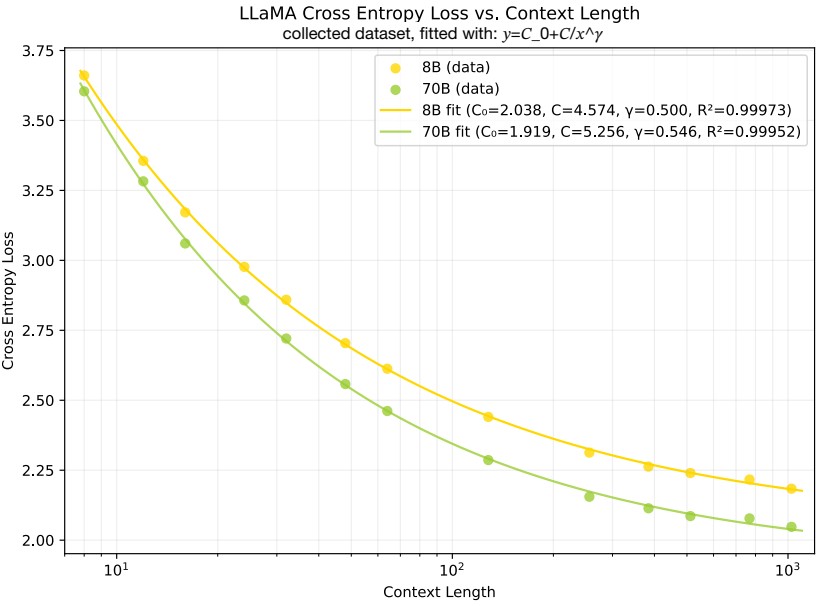

Figure 10: Cross Entropy Loss vs. Context Length, with log scale. We see that $y = C_0 + C/x^\gamma$ fits this curve well.

According to Figure 10, we see that $CE = C_0 + C/l^\gamma$ approximates well for text corpora that are sure not to be seen by the model.

## F Optimal Context Length for Two-needle-in-Haystack training: study on Synthetic Dataset

Here, we utilize our proposed synthetic dataset as a proxy to study the two-needle-in-haystack experiment (as we mentioned in Figure 5.)

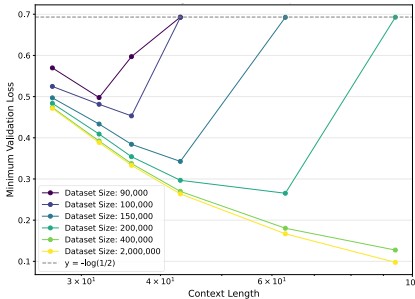

Figure 11: Validation set Cross Entropy Loss of the output token (resembling the 'answer' token of two-needle-in-haystack tasks) vs. context length, for MLPs trained with different training dataset sizes.

As mentioned in previous studies, the Cross Entropy loss of key tokens (e.g. the perplexity of the Answer token for Needles in Haystacks (NIH)) is highly correlated with downstream task accuracy (that is, the NIH tasks). Here, we use the Cross Entropy loss of the output bit of our synthetic task (shown in Figure 5) as a metric for our synthetic 'two-needles-in-haystack' task.

Here, we use a synthetic dataset similar to that mentioned in the main paper, except that it has more than 500 context bits (though most tasks require only first 100 context bits). In this section, we fix the size of the training data, train multiple iterations till overfitting, and take the best validation loss as the validation loss of that (training dataset size, context length) pair. Results are shown in **Figure 11**. From the result, we make such observation:

There exists an optimal context length for most training dataset size used, and such optimal context length increases with the amount of available training data.

This proves the concept that, when training dataset is limited, optimal context length smaller than the task length could potentially exists for tasks resembling the two-needle-in-haystacks tasks, and larger training dataset leads to larger optimal context length.

## G   EXPERIMENTS ON OTHER LANGUAGE MODELS

In our main paper (Figure 2), we present the relationship between Intrinsic Entropy and Cross Entropy Loss across multiple Language Models: Llama-3.1-8B, Qwen3-8B-Base, and RecurrentGemma-9B on OpenWebText.

For Qwen3-8B-Base, the linear relationship between Intrinsic Entropy and Cross Entropy loss holds quite well. For RecurrentGemma-9B, we observe that its Cross Entropy loss is significantly higher than Llama-3.1-8B and Qwen3-8B-Base for small context length (the 3 high points drawn on the figure), while other points show similar cross entropy loss. Therefore, we conclude that **RecurrentGemma-9B is not a good approximation for Bayes Model for these outlier points** (**i.e. it can't model low-context quite well with Cross Entropy loss** $> 5$, potentially because of its architecture or training pipeline), and we use the rest points where it is closer to Llama-3.1-8B and Qwen3-8B-Base as Bayes Model for regression.

Experiment in this section proves that, (1) our proposed Intrinsic Entropy and Cross Entropy loss relationship **holds across different series of Language Models with different architectures** when they are well-trained and can represent Bayes Models; and (2) the discovered relationship **only holds when the measured Language Model approximates Bayes Model well**.

## H   EIGENVALUE-BASED INTRINSIC ENTROPY ESTIMATION

In the main paper, we use Gaussian-KDE to estimate Intrinsic Entropy. Here we present an alternative eigenvalue-based estimation method, and show that both methods yield consistent linear relationships between Intrinsic Entropy and Cross Entropy loss.

## H.1 BAYES RISK VS. CONTEXT LENGTH ON OPENWEBTEXT

We use well-trained Large Language Models to approximate the Bayes Risk $H(P_l)$ on OpenWeb-Text. As shown in Figure 12, the relationship $H(P, P_l) \approx C_0 + C/l^{\gamma}$ (Equation 3) approximates the experimented behavior on OpenWebText well.

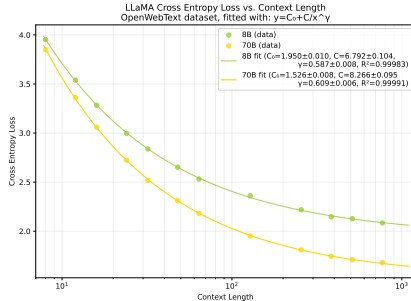

Figure 12: **Bayes Risk vs. Context Length**: Bayes Risk is approximated by Cross Entropy loss measured with LLaMa-3.1 series on OpenWebText, for different context length.

## H.2 EIGENVALUE-BASED ENTROPY MEASUREMENT

To establish a relationship between Cross Entropy and Intrinsic Space, we run LLaMa-3-8b on a subset of the Openwebtext dataset and obtain the feature of the last token as the feature representation, or Intrinsic Space of the approximated Bayes Model. For certain context length, we gather the feature representation of multiple ($\geq 10000$) samples, and conduct PCA analysis on these samples to obtain eigenvalues for the specific context length, results are presented in Figure 14. We see that the model with larger context length tends to have larger relative eigenvalues in intrinsic space, thus containing more information.

According to Statistical Physics, entropy of a system can be defined as $S = \log \Omega$ where $\Omega$ is the possible number of states of the system (Landau & Lifshitz, 1980). Similarly, we use the sum of logarithm of eigen values as proxy for measuring Information Entropy:[7]

$$
\begin{aligned}
S &= \log \Omega \\
&= \log V/h^{dim(V)} \text{ where } V \text{ is the volume in intrinsic space} \\
&= \sum_i \log rel\_eigval_i/h \\
&= \sum_i \log rel\_eigval_i + Const
\end{aligned}
$$

Here $h$ is the 'plank constant', meaning that one state corresponds to a unit hyper-volume of $h^{dim}$ in the Intrinsic Space. A different value of $h$ would only add a constant to $S$ and would not affect change in Entropy. Thus, we use $\sum_i \log rel\_eigval_i$ as Entropy in Intrinsic Space.

## H.3 CE LOSS VS. EIGENVALUE-BASED INTRINSIC ENTROPY

Experiments show that, no matter what subspace we use, the Cross Entropy Loss usually follows a linear relationship with the Entropy we measured in the subspace, **supporting the claim that the next token prediction task likely lies in some subspace of the Intrinsic Space, or (statistically) its Entropy should be some weighted average of Entropy of several subspaces of similar dimension.**. This also suggests that $H_{ntp}$ is approximately linear with $H_{IS}$, which validates our previous assumptions and claims.

---

[7]Similar estimation can also be derived from the assumption of Gaussian differential entropy with homogeneous reference measure.

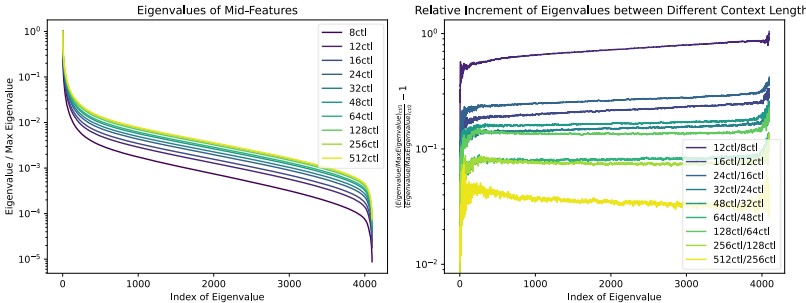

Figure 13: **Left**: **Relative Eigen Value** Measured for the last token, for LLaMa-3.1-8B on a subset of OpenWebText. **Right**: relative increment of relative eigenvalues (for different context lengths measured). We can see that the relative eigenvalues approximately increase at a same scale.

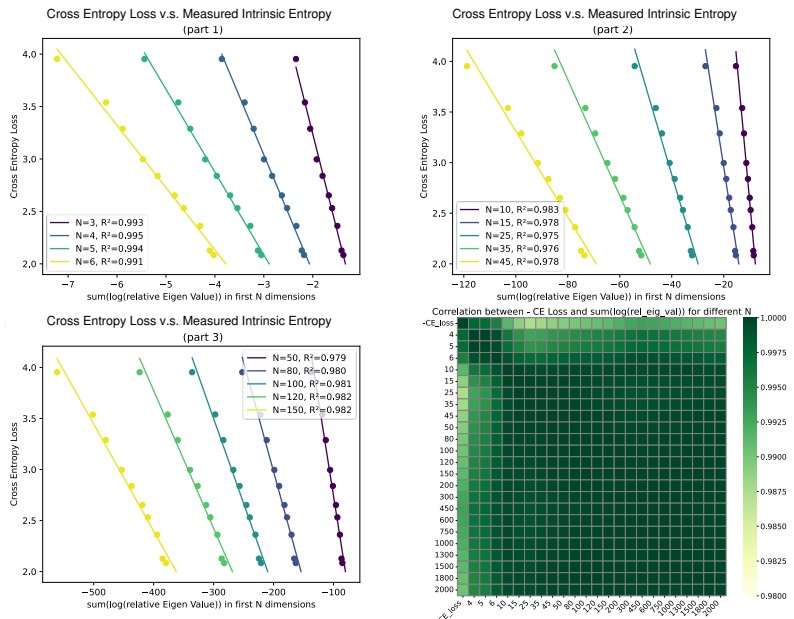

Figure 14: Upper-left, Upper-right, Bottom-left: **Cross Entropy Loss** vs. **measured Intrinsic Entropy** with $N$ first Eigen Values: $\sum_{i \leq N} \log$ rel_eig_val; Bottom-right: correlation between minus CE loss and $\sum_{i \leq N} \log$ rel_eig_val. All experiments are for LLaMa-3.1-8B on a subset of Open-WebText. From the first three figure, we see CE loss is linear with the Entropy of certain subspaces. From the bottom-right figure, we see that Entropy measured in different subspaces are highly correlated ($corr > 0.97$), which are also highly correlated with the CE loss for Next Token Prediction.

We observe a fairly linear relationship between CE Loss and Entropy measured (supporting our theory), validating our theoretical assumptions:

$$R_{Bayes} \approx -k * S(P_l) + Const,$$

which aligns well with Equation 2, thus validating our entropy-based deduction.

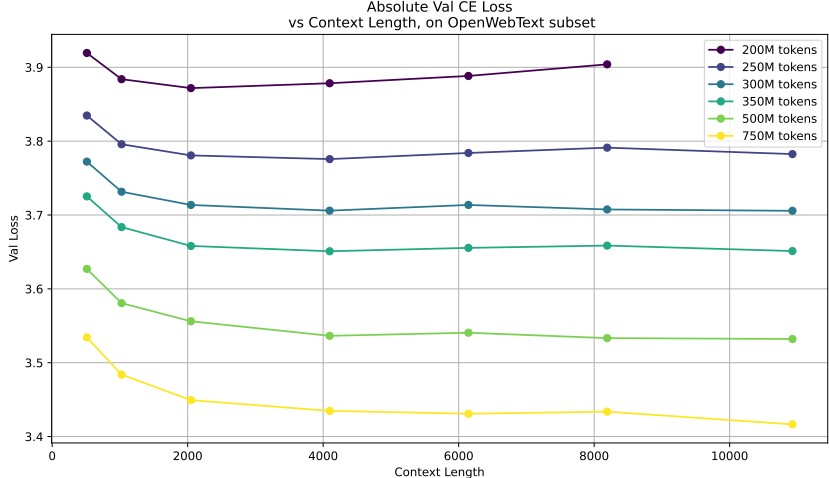

Figure 15: **Openwebtext subset**, Validation Loss vs. Context Length, for different dataset sizes. Different curves represent different amount of training data used. A more readable figure can be found in Figure 1, where the minimum validation loss reachable for each training dataset size is subtracted.

## I   EXPERIMENT SETTINGS

### I.1   NATURAL LANGUAGE DATA

#### I.1.1   OPTIMAL CONTEXT LENGTH EXPERIMENTS

As described in Section 3, we use nanogpt (Karpathy, 2022) and train a model with GPT-2 (Radford et al., 2019) architecture on a subset of OpenWebText dataset. For training, we use the AdamW (Loshchilov & Hutter, 2019) optimizer, learning rate of $6e-4$, weight decay of $1e-1$, 1000 warm-up iterations. For given token number, all models with different context length are trained with same number of iterations, where iteration number equals roughly to $token\_number/(0.1M)$.

We first select text corpora with context length beyond specific limits larger than the maximum training context length from OpenWebText, then split into Training set and Validation Set. The validation set has $134M$ tokens.

Experiments presented in Figure 1 and Figure 15 took around 300 gpu hours on 8 AMD MI-250X GPUs (which are similar in performance to Nvidia A100 gpus).

#### I.1.2   INTRINSIC DIMENSION EXPERIMENTS

We select long enough text corpora from the Openwebtext dataset. Then, following previous practice (Cheng et al., 2023), we conduct experiments with LLaMa-3.1-8b on 10000 samples of this subset. We extract the feature representation of the last token in the last layer, as the Intrinsic Representation of samples.

Conducting all intrinsic dimension measurements cost up to around 100 gpu hours for MI-250X gpus.

## J   RELATED WORK

### J.1   ENLARGING CONTEXT LENGTH FOR LMS

Previous work has made attempts to enlarge the context length of Language Models. Work represented by RoPE (Su et al., 2023) uses rotary positional embedding to support generalizing LMs to

longer context in inference compared to the training process. These work use modified positional embeddings to model the relative position dependency in attention mechanism.

There is also work about enhancing long context understanding and exploring Scaling Laws for context length (Xiong et al., 2024). These work utilize an adjusted pretraining and instruction-finetuning process with more long-context data to enhance the models' ability on long contexts.

Other work modifying architectures has also been proposed to enhance long context modeling or to simplify deployment of long context LLMs. For example, (Tang et al., 2024) proposes a training-free RazorAttention algorithm to largely compress the KV cache while maintaining performance unchanged.

Architectures and inference methods have been proposed to reduce inference time and memory cost for Language Models, represented by a series of linear transformer or RNN-based methods (Katharopoulos et al., 2020; Gu & Dao, 2024; Sun et al., 2024). These methods, largely reducing the computational cost and memory usage for long input contexts, have displayed a margin ahead of traditional attention-based (Lan et al., 2025; Li et al., 2025; Cai et al., 2025) language models for long context inference.

Currently a common practice to train very large Language Models supporting long context is to use pretrain the model with shorter contexts, then finetune them with longer contexts, as presented in tech reports of LLaMa-3 (Grattafiori et al., 2024) and DeepSeek-v3 (DeepSeek-AI et al., 2024).

## J.2 Irrelevant Long Context hurts performance of LMs

Besides context length scaling with relevant contexts, previous research have studied how LLMs perform for long irrelevant contexts. As an example, (Levy et al., 2024) studies the performance of current LLMs on an adjusted version of 'needle in a haystack task, where two pieces of key information are embedded into a long text corpora and a question related to both is asked, similar to that presented in Figure 5. The conclusion of these work is that LLMs would perform worse when there is too much irrelevant information.

## J.3 Long Context in another field: Time Series Forecasting

Context length, representing the length of input context, is not unique to Nature Language. For time series forecasting, where machine learning play an important row, there is also work discussing the impact of context length, represented by (Shi et al., 2024). These investigations find that there exists an optimal look-back horizon, which increases with dataset size. However, time series datasets are relatively small compared to NLP datasets, and thus whether this conclusion holds on NLP remains an open problem for this work to study.

## J.4 Related Theories for Scaling Laws

Since the discovery of Scaling Laws for Large Language Models (Kaplan et al., 2020) or even earlier, there has been theoretical work trying to explain why model performance could benefit from more data points and more model parameters. For example, (Sharma & Kaplan, 2022) studies the dataset and model scaling from the data manifold perspective.

Specially for Language Models, there is also previous work proposing all kinds of theoretical models. For example, (Michaud et al., 2024) proposes a feature-quant based theory; (Aghajanyan et al., 2020) views the effect of fine-tuning from the intrinsic dimension perspective; (Havrilla & Liao, 2024) proposes to understand scaling with intrinsic dimensions.

# K Intrinsic Dimension perspective: measurements in Intrinsic Space

## K.1 Bayes Risk from an Intrinsic Dimension perspective: Assumptions

Here we derive similar results as in Section 2.2, but from an Intrinsic Dimension perspective rather than an Information Entropy perspective.

We propose a simple theory model to relate $H(P, P_l)$ with the intrinsic dimension $dim(l)$ of the intrinsic space $space_l$ of the text corpora of length $l$ (for the next-token prediction task).

We assume these assumptions hold for Intrinsic Space (please see formal definitions of Intrinsic Space in Appendix D),

- Assumption 1. Intrinsic Dimension of the Bayes Model $\lim_{l \to \infty} dim(l) = dim(\infty)$ is finite, which is the Intrinsic Dimension of next token prediction of language itself.

- Assumption 2. $\forall l_1, l_2$ such that $l_1 < l_2$, $dim(l_1) < dim(l_2)$. This is because a longer context contains more information about the next possible token.

To simplify deduction, we further assume that,

- Assumption 3. **Uniform Information Gain** ($s$-bits for next token prediction per Intrinsic Dimension): Each intrinsic dimension would add $s$ bits of information to the next-token prediction task, so there are $dim(l) * s$ bits of information that can be represented in $space_l$ for the next-token prediction. This means the KL-divergence for the Bayes Model of context length $l$, $P_l$, with Bayes Model of infinite context length, $P = P_\infty$, is: $D_{KL}(P, P_l) = s * (dim(\infty) - dim(l))$. **Note this does not mean these are the only information in the Intrinsic Space, hence $s$ can be small, or even smaller than** $1$.

With these assumptions, we can derive $H(P, P_l)$ with $dim(l)$:

$$
\begin{aligned}
R_{Bayes} &= H(P, P_l) \\
&= -s * dim(l) + Const
\end{aligned}
\tag{34}
$$

This **linear relationship** can be observed in experiments for LMs and synthetic data, providing an alternative explanation to the entropy-based approach in the main paper.

## K.2 EXPERIMENTALLY MEASURE INTRINSIC DIMENSION USING PCA

We further use PCA as a metric to measure the Intrinsic Dimension of Dataset with respect to context length. We provide the relative degradation of the eigenvalue in the feature space of LLaMa-3.1-8B, for the last token. We see that larger input length would indeed provide feature with lower degradation in the intrinsic space. Notably, when $5 < idx < 1500$ the curves is similar to Zip-f distribution ($\log eig = C_0 - C * \log idx$), and for $500 < idx < 4000$ it resembles exponential degradation ($\log eig = C_0 - C * idx$).

Instead, following previous practice, here we use some **threshold** to decide the transformation index of these two states as Intrinsic Dimension: $\max_{idx} rela\_eig(idx) \geq$ threshold is used as the measured **Intrinsic Dimension**. Notably, the threshold here is a hyperparameter which is set to constants in previous work(e.g.1/20 in (Aghajanyan et al., 2020)), but we observe here that many thresholds would validate the linear correspondence of Cross Entropy vs. Intrinsic Dimension, which further enhance the robustness of our result. We use thresholds from $0.002$ to $0.25$.

For a certain threshold, we conduct experiments on several context lengths, and measure CE Loss on certain text corpora with these context lengths. We observe a fairly linear relationship between CE Loss and ID measured (supporting our theory), as shown in Figure 16. We see that, no matter what threshold we use, the Cross Entropy Loss usually follows a linear relationship with the Intrinsic Dimension we measured, showing the robustness of the PCA evaluation method, and validating our theoretical assumptions:

$$
R_{Bayes} \approx -s * dim(l) + Const,
$$

which aligns well with **Equation 34**, thus validating our intrinsic dimension-based deduction.

## K.3 MLP-BASED SYNTHETIC DATASET: INTRINSIC DIMENSION EXPERIMENTS

We train a large enough MLP on data generated on the synthetic tasks, and evaluate our model on the validation dataset. We train until overfitting the training dataset. We assume 1 dimension in

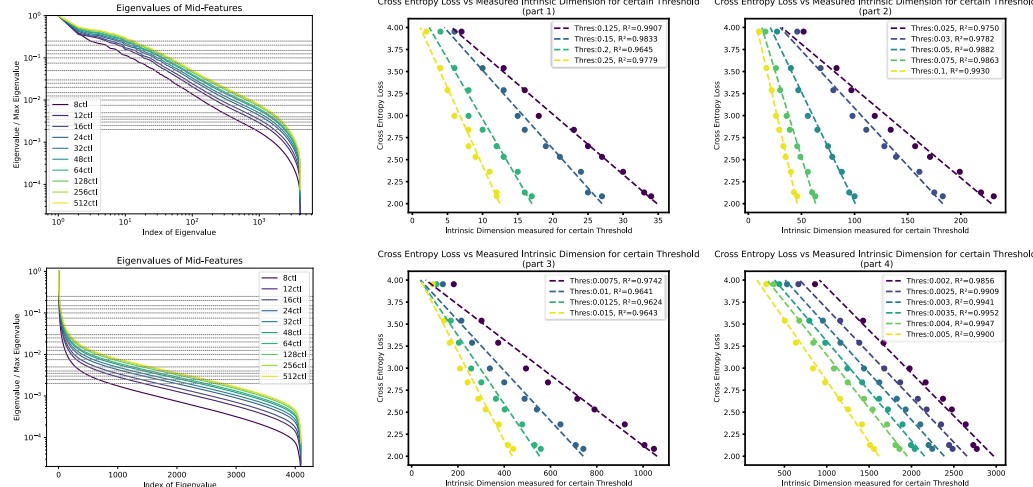

Figure 16: **Left figures**: **Relative Eigen Value** for LLaMa-3.1-8B on a subset of OpenWebText, presented in different x-axis scales, with different context length visible to Language Model. Gray lines represent different **threshold**s we take to measure the intrinsic dimension of the current model. **Right figures**: **Cross Entropy Loss** vs. **Measured Intrinsic Dimension**. Each line represents a certain threshold used to measure ID in the intrinsic space of the used LLM. Different Measurements would give ID values that are linear w.r.t. each other, and they are all linear w.r.t. CE loss.

Intrinsic Space can store information about 1 subtask, hence we take $ID(l) = t(l)$ as its theoretical value here.

Let $f(x, C, C_0, \gamma) = C_0 - C/x^\gamma$ and $g(x, k, b) = k * x + b$.

The fitted results are:

- ID & CL: $ID \approx f(CL, C, C_0, \gamma)$, $C_0 = 51.1 \pm 1.0$, $C = 1.7 * 10^3 \pm 0.3 * 10^3$, $\gamma = 1.18 \pm 0.06$, $R^2 = 0.9997$.

- CE & CL: $CE \approx f(CL, C, C_0, \gamma)$, $C_0 = -0.015 \pm 0.013$, $C = -23.8 \pm 4.3$, $\gamma = 1.18 \pm 0.06$, $R^2 = 0.9997$.

- CE & ID: $CE \approx g(ID, k, b)$, $k = -0.693 \pm 1 * 10^{-5}$, $b \approx 0.0139 \pm 4 * 10^{-7}$, $R^2 = 1 - 7 * 10^{-9}$.

As shown, we construct synthetic data such that $ID(l) = ID_0 - C'/l^\gamma$, and our measurements show $CE = C + C'/l^\gamma$. More importantly, **for the synthetic data example, Cross Entropy loss is almost perfectly linear with the Intrinsic Dimension as we defined previously.** This validates the linear relationship between Cross Entropy Loss and Intrinsic Dimension; and we have also provided a construction to match the measured relationship $CE(l) \approx C_0 + C/l^\gamma$.

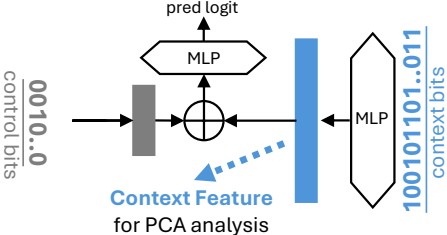

Figure 17: Model trained on the proposed synthetic dataset; $\oplus$ represents feature concatenation. Only the first $l$ bits are used as input to context MLP when the context length is set to $l$. We conduct PCA on Context Feature to analyze the intrinsic dimension of input context bits for various context lengths.

We train a model with a specialized architecture, allowing us to use the feature representation of a middle layer as a feature vector for input context bits, as shown in **Figure 17**. After training the model on data with different context length, we conduct PCA on the obtain context feature representation to study the Intrinsic Space of this model.

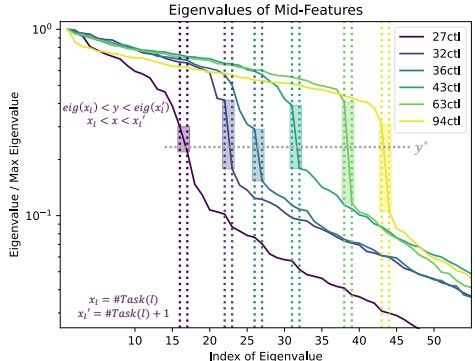

Figure 18: Relative eigen value vs. index, for models trained on different context length. Vertical lines: $x_l = ID(l)$ and $x'_l = ID(l+1)$. For example, a context length 27 has 16 subtasks visible, corresponding to 16 bits in Intrinsic Space. Assuming 1 dimension in Intrinsic Space represents 1 bit, the leftmost purple rectangle drawn means a range of $y_{threshold}$ that would provide an accurate estimation of $ID(27) = 16$ for context length 27. There exists $y^*$ that would provide an estimation of $ID$ for all context lengths, as shown in the figure.

We find that: (1) the neural network would indeed learn the key information in the context bits. For models with different input context lengths, although their inner dimensions are the same (80), the representation of inputs in this inner space mainly lies in the first $ID$ dimensions, and the eigen values corresponding to other dimensions are very small; and (2) there exists such threshold $y^*$ that would estimate $ID$ for all context lengths accurately. We can take some threshold $y^*$ to estimate the intrinsic dimension, by obtaining the maximum index of the relative eigen value such that the relative eigen value is larger than $y^*$, which would give accurate and consistent estimates.

### K.4 BRIDGING THE GAP BETWEEN INTRINSIC DIMENSION EXPLANATION AND INTRINSIC ENTROPY EXPLANATION

Here, starting from previous assumptions and measurements w.r.t. Entropy in Intrinsic Space, we explain why CE is linear w.r.t. Intrinsic Dimension measured in Section 2.2, for $idx > 500$. We see in Figure 13 that for $idx > 500$, the relative eigenvalues mainly follow an exponential decay:

$$releigval_{l,\,idx} = releigval_{l,\,0} * \exp\{-\alpha_l * idx\}, \text{ for certain context length}$$

where $l$ is the context length, $idx$ is the index of some certain eigen value, and $\alpha_l$ is the exponential decay coefficient for this certain context length $l$.

We also see from the previous results (**Figure 13**) that for different context lengths, the relative eigenvalues increase almost in the same proportion, especially for $idx > 1000$. That is, $\alpha_l \approx \alpha$. We define $\gamma(l) = releigval_{l,0}/releigval_{\infty,0}$ and thus we have: $releigval_{l,idx} = religval_{\infty,0} * \gamma(l) * \exp(-\alpha * idx)$.

For the subspace for the next token prediction task, we denote its dimension to be $m$. Hence, the entropy should be proportional to log of volume in the subspace; that is:

$$
\begin{aligned}
S_{subspace}(l) &= \sum_{idx \in \{\text{dimension of subspace}\}} \log releigval_{\infty,0}\gamma(l)\exp(-\alpha * idx) \\
&= m\log\gamma(l) + Const
\end{aligned}
\tag{35}
$$

which is the result of the **Intrinsic Entropy Explanation**.

For **Intrinsic Dimension explanation**, if we are using some certain threshold $thres$ to measure Intrinsc Dimension, the measured dimension $dim(l, thres)$ should satisfy:

$$releigval_{\infty,0} * \gamma(l) * \exp\{-\alpha * dim(l, thres)\} = thres,$$

hence the measured dimension is $dim(l, thres) = 1/\alpha * (\log \gamma(l) + \log(releigval_{\infty,0}/thres))$. Plugging this into Equation (35) we have:

$$L_{CE} = -S_{subspace}(l) + Const = -m\alpha * dim(l, thres) + Const(thres). \tag{36}$$

Thus, we derive our assumptions in Section 2, where $s = m\alpha$. Equation (36) can also be validated in the lower-right part of Figure 16, where the Intrinsic Dimensions (for $idx \geq 500$) are measured in the exponential decay area, and these lines, though measured with different threshold ($thres$), share similar slopes w.r.t. CE Loss (that is not related with threshold, as shown in Equation 36).

## L    DISCLOSURE OF LLM USAGE

LLMs are used in this work for polishing writing only.

