# OpenReview forum: "Intrinsic Entropy of Context Length Scaling in LLMs"
_ICLR.cc/2026/Conference — ICLR 2026 Oral_

### Official Review · Reviewer_ULtS · 2025-10-22

**Soundness:** 3
**Presentation:** 3
**Contribution:** 3
**Rating:** 6
**Confidence:** 3

**Summary:**

The paper introduces a theoretical and empirical framework for
understanding how context length affects the performance and scaling
behavior of large language models. The authors propose the concept of
Intrinsic Entropy, which quantifies the information a model can encode
within its internal (“intrinsic”) representation space given a
certain context length. They decompose cross-entropy loss into two
parts — the Bayes Risk, representing the irreducible uncertainty from
limited context, and the Approximation Loss, representing imperfect
model learning. From this perspective, the Bayes Risk decreases
monotonically with longer context since more information is available,
whereas the Approximation Loss tends to increase because longer
contexts enlarge the model’s effective input dimension and complicate
optimization. The trade-off between these opposing effects gives rise
to an optimal context length that minimizes total loss for a given
dataset size.

Empirically, experiments using LLaMA-3.1 (8B and 70B) on OpenWebText
subsets and a synthetic parity dataset confirm the theory. The
measured relationship between Intrinsic Entropy and Cross-Entropy Loss
is nearly linear (correlation > 0.97), and the validation loss across
context lengths follows a scaling law of the form H=C0 + C / l^gamma.
. Moreover, when varying training-data size, the optimal context
length shifts upward — indicating that larger datasets justify
proportionally longer contexts. The findings imply that excessively
extending context windows can degrade performance when data are
insufficient, emphasizing the need for joint optimization of context
length, data size, and model capacity. This framework bridges
statistical physics–style scaling laws with information-theoretic
reasoning, offering a principled explanation for when and why long
contexts help or harm LLMs.

**Strengths:**

1. The paper introduces a novel concept — Intrinsic Entropy — that
links information-theoretic measures to model-internal representations
(“intrinsic space”). This provides a principled bridge between
entropy, Bayes risk, and approximation loss, enabling a quantitative
explanation for the existence of an optimal context length. By
deriving analytical relationships the work extends traditional
scaling-law theory (e.g., Kaplan et al. 2020) to the underexplored
dimension of context length. This is conceptually powerful: it
reframes the intuition that “longer is better” into a mathematically
grounded trade-off between information gain and representational
complexity.




2. Experimentally, the paper validates its theory using LLaMA 3.1 models
and synthetic data, showing a near-perfect linear correlation (R^2 >
0.97) between Intrinsic Entropy and Cross-Entropy loss, across varying
context lengths. The authors further identify a data-dependent optimal
context length, which scales predictably with dataset size, and
reproduce this result across different datasets. This dual
theoretical–empirical consistency gives the framework strong
credibility and practical relevance: it offers a diagnostic tool for
optimizing context length in training regimes and provides insight
into the physics-like scaling behavior of LLMs.  In short, the paper’s strength lies in its integration of
first-principles reasoning, strong empirical validation, and broad
applicability — offering both a new theoretical lens and actionable
insight into how LLMs handle long contexts.

**Weaknesses:**

1. The paper’s theory hinges on several idealized assumptions that may
not hold in practice: The notion of a stable “Intrinsic Space” — a well-behaved latent
manifold shared across models — is assumed rather than
demonstrated. In reality, representations in transformer layers are
non-stationary and architecture-dependent, making it difficult to
assert that a single intrinsic space can be defined across context
lengths. The linearity assumptions — e.g., that Bayes Risk is linearly
proportional to Intrinsic Entropy or Intrinsic Dimension — simplify
the analysis but are only loosely supported empirically. These
relationships might hold approximately in narrow regimes but could
break under different architectures, data modalities, or tokenization
schemes. While mathematically elegant, the framework risks being too abstract
to describe the heterogeneous behaviors of real LLMs under diverse
conditions.



2. The experiments, though carefully designed, rely on a small number of
datasets and model families: Only LLaMA 3.1 (8B and 70B) and OpenWebText subsets are tested for
natural language, with one synthetic dataset for controlled
validation. There is no downstream evaluation (e.g., reasoning or QA tasks) to
verify whether the proposed “optimal context length” correlates with
functional performance. The paper also lacks cross-model validation — results are not shown
for alternative architectures like Mistral or Gemini-style
mixture-of-experts, which could challenge the universality of the
scaling law. The findings, while compelling within the tested settings, remain
tentative as a general principle across model types and data regimes.



3. Although the “Intrinsic Entropy” construct is theoretically neat, it
is not directly measurable without heavy computational overhead (e.g.,
eigenvalue decompositions of hidden representations across
contexts). Moreover, its connection to optimization dynamics and
architecture-level design remains abstract — the paper suggests
guidelines for tuning context length but provides no concrete
algorithm or scaling heuristic to implement this in real-world
training. The work’s insights are conceptually illuminating but operationally
difficult to apply for practitioners designing next-generation
long-context LLMs.

**Questions:**

1. The entire theory rests on the existence of a stable Intrinsic Space
where internal representations of data reside and where entropy can be
measured. However, as the authors themselves admit, this space may
depend not only on the data but also on the neural network
architecture and training dynamics.  Can Intrinsic Entropy be defined independently of specific
architectures or layer representations, making it a universal property
of language modeling itself?


2. While the framework explains cross-entropy scaling, it is unclear
whether higher or lower Intrinsic Entropy predicts better reasoning,
retrieval, or long-document comprehension. The authors validate on
synthetic and language data but do not link the proposed measure to
functional metrics. Does Intrinsic Entropy correlate with actual task-level performance,
or does it capture only internal statistical regularities?

3. Can the optimal context-length principle generalize across models,
data types, and modalities? The observed law that the optimal context length increases with
dataset size is based on a limited set of models (LLaMA-3.1) and
corpora (OpenWebText, synthetic parity). Whether this scaling
relationship extends to multimodal LLMs, retrieval-augmented
architectures, or dialogue models remains untested.  Is the “optimal context length” phenomenon universal across model
families and data modalities, or an artifact of specific transformer
training regimes?

---

> ### Author Response · Authors · 2025-11-21
>
> We would like to thank the reviewer for recognizing our strength in introducing the novel concept of Intrinsic Entropy to link information measurements to model-internal representation, Bayes Risk and Approximation Loss, and our empirical discoveries on NLP datasets and synthetic datasets as well. We would like to address your concerns bellow,
>
> ## `Intrinsic Entropy across Model architecture`
> We further conduct experiments using other model architectures, experiments of which are shown in appendix G of the refined paper.W
> We conduct similar experiments on Llama-3.1-8B, Qwen-3-8B-Base, and Recurrent-Gemma-9B. As shown in **Figure 13 in Appendix G in our revised paper**, when the models are close to Bayes Model, the relationship between Cross Entropy Loss and Intrinsic Entropy is similar across different family of Language Models trained or their architecture used.
>
> ## `Intrinsic Entropy and Down-stream tasks`
>
> **We further conduct analysis for Intrinsic Entropy and optimal context length on downstream tasks**. Please refer to the following contents:
> ### `RULER-QA1 dataset:Optimal context length still exists`
>
> Our conclusion on downstream task is:
>
> 1. There also exists an “optimal context length”
>    On the task we tested, such “optimal context length” increases when the task requires long context.
> 2. When context length is small and the model has not yet deviated from “Bayes Model”, Intrinsic Entropy is positively-correlated with QA accuracy.
>
> To study whether “Optimal Context Length” also exists for down-stream tasks, we study Llama-3.1-8B’s performance on a modified RULER-QA1[1] dataset, as added in **Appendix A in our revised paper**.
>
> The RULER-QA1 dataset is constructed by extending standard document-QA tasks through inserting the “golden paragraph” (paragraphs containing answers) uniformly into other paragraphs from the same SQuAD dataset. Our modified, Position-weighted-RULER-QA1 modifies the distribution of “golden paragraph” from uniform distribution into $P(x)\propto (1-x/L)^\gamma$, where $x$ is the distance (in token) of the “golden paragraph” to the end of input, and $L$ is the length (in token) of the concatenated sample, gamma is a hyper-parameter to control how close the golden paragraphs are distributed to end of input. (larger $\gamma$ means less long-context dependency, while $\gamma=0$ is uniform distribution).
>
> From **Figure 9 in Appendix A in our revised paper**, we have these observations:
>
> 1. For any $\gamma$, when increasing model’s context length, the QA accuracy first improves mainly because the model can see more “golden paragraphs” (i.e. Bayes Risk decrements faster than Approximation Loss increment); then it drops because the model becomes unable to handle longer contexts (i.e. increment in Approximation Loss surpasses the decrement of Bayes Risk). Therefore, there is still an optimal context length, for certain down-stream tasks.
> 2. For smaller $\gamma$ (i.e. the task depends more on tokens farther away), the optimal context length is larger. This aligns with the intuition that, for applications that require larger typical context length, the required “optimal context length” would be larger.
> 3. When increasing context length, before QA accuracy starts to decrease (i.e. model deviates from Bayes Model), **the Intrinsic Entropy still increases** along with QA accuracy.
>
> We provide an intuitive analysis from loss decomposition perspective. Intuitively, some tasks require larger context lengths to solve (i.e. the Bayes Risk of that metric decreases slower with context length compared to other tasks like Cross Entropy loss for next token prediction), thus they need more contexts. However, since the model’s performance would decrease for long contexts after all (i.e. the Approximation Loss still increases with context length), the balance of these two terms still leads to an optimal context length. In our original main paper, we fixed task, vary training set size, and show optimal context length increases with training budget and dataset size; in this supplemented experiment, we fix the model (so we fix training budget and dataset as well), and show optimal context length increases with (downstream) tasks’ typical context length.

---

> > ### Author Response · Authors · 2025-11-21
> >
> > ### `Synthetic Data: proof of concept`
> > As presented in Figure 5, in our original main paper, our synthetic dataset is inspired by the two-needle-in-haystacks task. Therefore, the cross entropy loss of its output token could be viewed as highly-correlated with “recall accuracy” for this synthetic two-needle-in-haystack task. We have proved that for the synthetic dataset, Intrinsic Entropy is linear with the Cross Entropy Loss of the output token. We further train a simple MLP on a similar synthetic task. This task has more context bits. We limit the horizon of the model to simulate a limited “context length”. We train models of each context length on a training dataset with certain sizes till overfitting and obtain the best validation loss.
> >
> > As shown, our result (as shown in **Appendix F Figure 12** in our new paper) also supports the claim that, for limited training dataset sizes, the balance between Bayes Risk (preferring longer context length) and Approximation Loss (preferring shorter context length) would result in an optimal context length (that increases with available training dataset size).
> >
> > Overall, we hope our further comments are helpful for your evaluation. We would like to thank the reviewer for the review.
> >
> > **P.S.**: We have uploaded our revised paper, where we use $\\color{blue}{\\text{blue texts}}$ to highlight updates.
> >
> > **Reference**:
> >
> > [1] What is Wrong with Perplexity for Long-context Language Modeling? COLM 2024 https://arxiv.org/abs/2404.06654

---

> > > ### Comment · Reviewer_ULtS · 2025-11-21
> > > **Thank you for your comment**
> > >
> > > Thank you for your reply. Your comments clarified my concerns. I'll update my scores after seeing all other discussions.

---

> ### Author Response · Authors · 2025-11-26
>
> We would like to thank the reviewer for the positive comment on our introducing the concept of Intrinsic Entropy for connecting information measurements to Bayes Risk, the discovery of the optimal concept length for language models, and our experiments on NLP datasets and synthetic datasets as well. Also, the suggestions on further experiments on downstream tasks and other language models are constructive and have helped us to improve the soundness of our work. Thank you again for your feedback!

---

### Official Review · Reviewer_vDBt · 2025-10-28

**Soundness:** 4
**Presentation:** 4
**Contribution:** 3
**Rating:** 10
**Confidence:** 4

**Summary:**

The authors propose a new scaling law that takes context length into account. It has the form of
$$L = C_0 + C/l^\gamma + A(l)/D^{\alpha(l)}$$
Where l is the context length. This form is a natural generalization of the scaling law suggested by Kaplan et. al. to the case of a variable context length. It captures the trade-off between reduced training sample efficiency and increased generation confidence that LLM experiences as the context length grows.

The authors study the form of $\alpha(l)$ and show that it can be represented as $\alpha(l) = c\dim(l)$ with $dim(l)$ being the dimensionality of the latent space necessary to represent text of length l. Since larger text naturally may contain more information, the latent dimensionality grows with l, which means that $\alpha(l)$ decreases. Thus, the suggested scaling law consists of two terms, one of which increases with l while the other one decreases, which implies the existence of an optimal value of l.

The results summarized here are shown rigorously in the presented work. It is worth noticing that the relation between the coefficient $\alpha$ of the scaling law and the intrinsic dimension of the data was known in the literature. The authors push it further by leveraging the relationship between the intrinsic dimension and the context length.

**Strengths:**

The paper presents an interesting result, consisting of a non-trivial observation and a sound and consistent theory.

The authors provide a convincing set of evidence that supports their theoretical assumptions.

**Weaknesses:**

The paper claims that the theoretical framework presented can provide practical insights, such as establishing that the size of the training dataset dictates an optimal context length and bounds context-length scaling. It seems to miss that, in practice, the context length is determined not to minimize the loss function but rather for other considerations, such as the typical context length in the target use cases and the capabilities of the computational hardware. If the prescribed “optimal” context length is larger than the typical context length in applications, the context length will still be kept within the application's demands due to computational constraints. If the prescribed “optimal” context length is shorter than the typical context length in applications, the context length will still be kept up to the application's demand, or the system would not be deployable.

Another argument against the practicality of the delivered result is that the real objective behind LLM training is not the cross-entropy values, but the evaluation results on downstream metrics. To prove the practicality of the results, the authors would need to conduct experiments that hold compute/memory constant and study task metrics (e.g., RAG accuracy, long-form QA F1, code completion pass@k) against context length

**Questions:**

NA

---

> ### Author Response · Authors · 2025-11-21
>
> We sincerely thank the reviewer for positive comments on our observations regarding the optimal context length, and the related theory and experiment analysis. We also appreciate your insightful feedback on the downstream tasks. We do recognize that, in practice, downstream tasks are important and might have different “typical context length”. Here, **we conduct analysis from a Doc-QA downstream task for Nature Language, and on a synthetic task:**
>
> ### `RULER-QA1 dataset:Optimal context length still exists`
>
> Our conclusion on downstream task is:
>
> 1. There also exists an “optimal context length”
>    On the task we tested, such “optimal context length” increases when the task requires long context.
> 2. When context length is small and the model has not yet deviated from “Bayes Model”, Intrinsic Entropy is positively-correlated with QA accuracy.
>
> To study whether “Optimal Context Length” also exists for down-stream tasks, we study Llama-3.1-8B’s performance on a modified RULER-QA1[1] dataset, as added in **Appendix A in our revised paper**.
>
> The RULER-QA1 dataset is constructed by extending standard document-QA tasks through inserting the “golden paragraph” (paragraphs containing answers) uniformly into other paragraphs from the same SQuAD dataset. Our modified, Position-weighted-RULER-QA1 modifies the distribution of “golden paragraph” from uniform distribution into $P(x)\propto (1-x/L)^\gamma$, where $x$ is the distance (in token) of the “golden paragraph” to the end of input, and $L$ is the length (in token) of the concatenated sample, gamma is a hyper-parameter to control how close the golden paragraphs are distributed to end of input. (larger $\gamma$ means less long-context dependency, while $\gamma=0$ is uniform distribution).
>
> From **Figure 9 in Appendix A in our revised paper**, we have these observations:
>
> 1. For any $\gamma$, when increasing model’s context length, the QA accuracy first improves mainly because the model can see more “golden paragraphs” (i.e. Bayes Risk decrements faster than Approximation Loss increment); then it drops because the model becomes unable to handle longer contexts (i.e. increment in Approximation Loss surpasses the decrement of Bayes Risk). Therefore, there is still an optimal context length, for certain down-stream tasks.
> 2. For smaller $\gamma$ (i.e. the task depends more on tokens farther away), the optimal context length is larger. This aligns with the intuition that, for applications that require larger typical context length, the required “optimal context length” would be larger.
> 3. When increasing context length, before QA accuracy starts to decrease (i.e. model deviates from Bayes Model), the Intrinsic Entropy still increases along with QA accuracy.
>
> We provide an intuitive analysis from loss decomposition perspective. Intuitively, some tasks require larger context lengths to solve (i.e. the Bayes Risk of that metric decreases slower with context length compared to other tasks like Cross Entropy loss for next token prediction), thus they need more contexts. However, since the model’s performance would decrease for long contexts after all (i.e. the Approximation Loss still increases with context length), the balance of these two terms still leads to an optimal context length. In our original main paper, we fixed task, vary training set size, and show optimal context length increases with training budget and dataset size; in this supplemented experiment, we fix the model (so we fix training budget and dataset as well), and show optimal context length increases with (downstream) tasks’ typical context length.

---

> > ### Author Response · Authors · 2025-11-21
> >
> > ### `Synthetic Dataset resembling two-needles-in-haystack task: optimal context length on "synthetic" retrieval task`
> >
> > In our original main paper, our synthetic dataset is inspired by the two-needle-in-haystacks task (as we presented in Figure 5). Therefore, the cross entropy loss of its output token could be viewed as highly-correlated with “recall accuracy” for this synthetic two-needle-in-haystack task. We have proved that for the synthetic dataset, Intrinsic Entropy is linear with the Cross Entropy Loss of the output token.
> >
> > For simplicity, we further train a simple MLP on a similar synthetic task. This task has more context bits. We limit the horizon of the model to simulate a limited “context length”. We train models of each context length on a training dataset with certain sizes till overfitting and obtain the best validation loss.
> >
> > As shown, our result (as shown in **Appendix F Figure 12 in our revised paper**) also supports the claim that, for limited training dataset sizes, the balance between Bayes Risk (preferring longer context length) and Approximation Loss (preferring shorter context length) would result in an optimal context length (that increases with available training dataset size).
> >
> > **Overall**, We would like to thank you for your appreciation in our theoretical and experimental discoveries and contributions, and your valuable suggestions with respect to insights on further tasks as well. We hope our further comments could be helpful for your evaluation.
> >
> > **P.S.**: We have uploaded our revised paper, where we use $\\color{blue}{\\text{blue texts}}$ to highlight updates.
> >
> > **Reference**:
> >
> > [1] What is Wrong with Perplexity for Long-context Language Modeling? COLM 2024 https://arxiv.org/abs/2404.06654

---

### Official Review · Reviewer_Mnty · 2025-10-29

**Soundness:** 1
**Presentation:** 1
**Contribution:** 2
**Rating:** 2
**Confidence:** 3

**Summary:**

The paper proposes to investigates into LLM context length from model entropy perspective. The paper leverages cross entropy (and treated as Bayes Risk in the paper, for example in Figure 2 and explained in 2.2.2) and show that it empirically bears a linear relationship with intrisic entropy which is calculated as log of the eigenvalues. Furthermore, by approxmimating the model approximation error in section 2.3, the paper shows theoretically that there exists an optimal context length in section 3 and also demonstrate it empirically.

**Strengths:**

The direction which contructs a decomposition and shows that optimally there exists an optimal context length as shown in section 3 is very interesting. Eq6/7 effectively shows that D would imply an optimal L. There are several empirical results that demonstrate this; besides, the linear relationship between cross entropy and intrinsic entropy seems novel and insightful.

**Weaknesses:**

I personally think that the mathematical construction lacks rigor. For example, there is no justification of equaling states with volume divided by dimensions. Although eq(3) is understandable, it is not an acceptable exposition since l is discrete. It is also problematic to directly treat crossentropy as Bayes Risk particularly there is no justification provided throughout the paper.

The paper is badly presented. The figures are not very readable most of the time with non informative titles (e.g. Figure 1, part1/part2/part3) and equations through 381-386. All these make papers more read as a draft instead of a formal conference submission. All the above significantly makes the reading process harder. For example, I am not able to understand the synthetic task presented in Figure 5 right part (with carefully reading its descriptions 372-377)/

**Questions:**

Figure 1 and Figure 7 shows the correlation between cross entropy and N. How are samples obtained for a particular N please? More in detail, how is correlation calculated for a particular (N, cross entropy) pair?

---

> ### Author Response · Authors · 2025-11-21
>
> We would like to thank the reviewer for finding it inspiring of the optimal context length, the experiment results and the novelty of Intrinsic Entropy. We also appreciate your feedback on presentation. We would like to address your questions and concerns below:
>
> ### `Regarding concerns of Intrinsic Entropy Estimation`
>
>
> We are glad to see the reviewer asking about the insightful similarity between our calculation of Intrinsic Entropy and Entropy as a physics quantity in Statistic Mechanics. Here are 3 aspects that we would like to clarify:
> 1. Our way to calculate Intrinsic Entropy (i.e. sum of log of eigenvalues), though inspired from the perspective of Physics Entropy, can also be derived from Gaussian differential entropy estimation.
> 2. We use the term “equaling states with volume” to geometrically describe a homogeneous reference measure in the Intrinsic Space: here it equivalently means homogeneous reference measure. Our intuition is to emphasize such assumed property of the Intrinsic Space, but we agree that adding the homogeneous reference measure perspective could be more rigorous, as we have added to the revised paper in **page 5, line 269 in our revised paper**.
> 3. In appendix B.2 in our original paper we use the improved Gaussian-KDE based methods for estimating Intrinsic Entropy, which also shows similar results.
>
>
>
> ### `Regarding concerns of taking derivative over context length, a discrete integer`
>
> **We follow previous work** where the notation $\partial f(x,\cdot)/\partial x$ where $x$ is a discrete value (e.g. parameter number, vocab size, etc.) has been widely used without explicitly pointing out: for example, in appendix B.1 in [1] ($\partial_N l|_C=0$ where $N$ is parameter number, $l$ is loss, $C$ is compute); in equation (7) in [2] ($\partial C/\partial V$ where $C$ is compute flops and $V$ is vocab size); etc. We follow these literature and view context length as a real value in $R$, and do not explicitly state how the derivatives are handled; however, this notation could be formally defined: the derivative can also be viewed as taking derivatives to functions $f_R(l_R)$ which is an extension of discrete function $f$. **We appreciate your feedback and agree that commenting this would add to math rigor**, hence we have added related comments in our refined paper; (e.g. please refer to line 213 in our refined paper).
>
> ### `Relationship between Cross Entropy Loss and Bayes Risk`
> In our paper, we use "Cross Entropy Loss" or $H(P,P_l)$ to denote the cross entropy between the true data distribution $P$ (natural language) and an optimal language model $P_l$​ with context length $l$. This is stated in our original paper around line 107, where we explicitly wrote that we are considering Bayes Risk-Approximation Loss decomposition for Cross Entropy, and pointed to Appendix B (Appendix A in original paper) for detailed definitions.This quantity represents the **minimum achievable loss** for any language model constrained to context length $l$. In decision-theoretic terms, this is precisely the **Bayes risk** for the language modeling problem under a cross-entropy loss function, where the model class is restricted to those using context length $l$.Therefore: The Bayes risk (with respect to cross-entropy loss for language models with context length $l$ equals the cross-entropy loss $H(P,P_l)$.
> For the experiment, we assume Llama-3-8B can handle contexts in Openwebtext well, and use it to “approximate” the optimal language model of context length $l$ for relatively small $l$, and we inspect the properties of it as a proxy of properties of optimal LM with context length $l$.
>
>
> ### `How Figure 1 and Figure 7 are conducted`
> As shown in **Section 2.2.2**, we are measuring the relationship between Cross Entropy Loss $CE$ and Intrinsic Entropy $IE$, for different context length $l$.
> In our main paper, we measure Intrinsic Entropy with the sum of log of top $N$ eigenvalues in the Intrinsic Space.
>
> As for the eigenvalues: For certain context length $L$, we sample $M>N$ samples from the Openwebtext dataset, feed them through the measured LM (e.g. Llama-3.1-8B), and obtain $M$ hidden vectors, each of shape $d$, which is the hidden dimension of the model. Next, we conduct PCA on such $M$ vectors of dimension $d$, and obtain top-$N$ eigenvalues.
>
> Figure 7 is obtained through a similar method on the synthetic dataset. In our revised paper, we have updated the caption of Figure 1 and add more descriptions to Section 2.2.2.
>
> ### `Others`
>
> We would like to thank you for your feedback when reading our paper. We have **revised captions and descriptions of pictures in our paper**, to make them more readable.
>
> **Overall**, we hope our comments will be helpful for your reading and evaluating our paper, and we hope our modifications will make our paper clearer to understand. Thanks again for your review.
>
> **P.S.**: We have uploaded our revised paper, where we use $\\color{blue}{\\text{blue texts}}$ to highlight updates.

---

> > ### Author Response · Authors · 2025-11-21
> >
> > **Reference**:
> >
> > [1] Scaling Laws for Neural Language Models, Openai and arxiv, https://arxiv.org/abs/2001.08361.
> >
> > [2] Scaling Laws with Vocabulary: Larger Models Deserve Larger Vocabularies, NeurIPS 24, https://arxiv.org/abs/2407.13623.

---

> ### Comment · Reviewer_Mnty · 2025-11-23
>
> Thanks the authors for clarifying my questions on the intrinsic entropy estimation, the mathematical formulation as well as the experimental settings (i.e. how figures are constructed). I have read the rebuttals as well as the revised sections and feel at myself that I understand the paper and the settings better.
>
> There are two problems for taking minimum achievable loss with the cross-entropy (apologies for not making this part clear in my very first review). First is the model might be overfitting (as it is part of training); second, as other reviewers pointed out as well, is the adequacy of perpexity itself as a measure. I have read from your other answers that you have also tested on QA which make the claim stronger. For me, given the scope and potential impact of the claim that the paper constructs, I would expect the measure to be taken for different language models of different capacity of skill to make the whole paper convincing.
>
> Finally, I am a bit confused on the paper presentation that may be considered esthetics. e.g. Why figure 1 has one figure that splits into 3 parts and then another figure? Besides, it is not cited anymore with cross reference in the text. Equation (3) is an interesting mathematical property but is not need to derive the optimal length claim so does not seem to be needed to be put at a central place. There might be a better way to expose the entropy approximation than just displaying Figure 2 centrally.
>
> Overall, as stated in my original overview: I found this paper potentially exciting. But the exposition problem (figure, put centrally only the main claims and organize the paper to center that) and further experiments to fully support its claim (which is definitely noticable) made me think that the paper should be further worked on to present to the community. I think it will be a paper of high quality then.
>
> I won't change my evaluation score but have lowerd by confidence score instead to reflect my situation described above (I have also increased soundness score and this is not a problem for the paper I think).

---

> ### Author Response · Authors · 2025-11-24
>
> Thanks for further comments. We would like to address these related concerns as follows:
>
> ### `Minimum achievable loss with cross entropy`
>
> **Training and Evaluating using the same metric is not necessarily 'overfitting'**
>
> We would like to clarify that training and evaluating using the same metric is not 'overfitting'. For example, to train and to evaluate with the same reward/return for RL agents, logistic regression trained and evaluated with CE loss.
> Language Models trained with "Cross Entropy Loss" does not necessarily mean that using "Cross Entropy Loss" as a metric would lead to "overfitting" issue.
>
> **Bayes Risk can be defined for all loss forms, and it is not related to specific trained model**
>
> We would also like to point out that Bayes Risk can be defined for all loss forms, and it is not related to specific training processes or specific neural network architecture. Here, it means the concept of the minimum loss achievable under certain limitations. As stated in Section 2 and Appendix B (or App. A in original paper), strictly speaking, here we define $R_{Bayes}(l)$ to be the Bayes-optimal Cross Entropy loss under a ($ctl\leq l$) constraint for Natural Language, where we think such definition is not related to overfitting issues. Also, we appreciate your feedback, and we would state this more clearly.
>
> ### `Further experiments on more tasks with more models conducted in rebuttal`
>
> We appreciate your feedback on more tasks and models. Actually we have also conducted experiments on QA tests and with other models as well. As stated in our other reply, please refer to **Appendix A in the revised paper** for experiments proving an optimal context length on QA tasks: here it shows the existence of optimal context length that increases with tasks' requirement on long context.
> Also, please refer to **Appendix G in the revised paper**, where we do more experiments on Qwen3-8B-Base and RecurrentGemma-9B, confirming the relationship between Cross Entropy Loss and Intrinsic Entropy for more models of different series and architecture.
>
> ### `Figure drawing and Presentation`
>
> We would like to thank you for noticing these; some of these are indeed for clarity of presentation. For example, for figure 1, it would be hard to draw all three sub-figures about CELoss v.s. sum_N(log(eigval)) into one figure since the slope of these lines could differ by more than $10$ times, and it won't look good to draw them in one figure. As for more details (e.g. presentation of Eq(3) in original paper and reference to figures), we further modified our paper to make the presentation better (also in $\\color{blue}{\\text{blue texts}}$); thanks for the feedback.
>
> **Overall**, we have been trying our best to optimize the presentation since long before submission, so as to make our work more readable. If there are still concepts or logics that might be hard to understand or may need further explanations, we will be happy for further discussions.

---

### Official Review · Reviewer_8bxz · 2025-10-30

**Soundness:** 2
**Presentation:** 3
**Contribution:** 2
**Rating:** 4
**Confidence:** 4

**Summary:**

This paper investigates the impact of context length on Language Model (LM) performance through a novel theoretical lens termed "Intrinsic Entropy." The authors posit that the cross-entropy loss can be decomposed into Bayes Risk (the irreducible error of an optimal model) and Approximation Loss (the error from the trained model failing to match the optimal model). The core contribution is the proposal that the Bayes Risk is linearly related to the Intrinsic Entropy of the data manifold, which itself is a function of context length. They further argue that while longer contexts reduce Bayes Risk by providing more information, they simultaneously increase the intrinsic dimension, making the Approximation Loss harder to minimize, especially with limited data. This trade-off leads to an "optimal context length" for a given training dataset size, beyond which validation loss increases. The paper supports its claims with experiments on both natural language (OpenWebText with LLaMa and GPT-2) and a carefully constructed synthetic dataset.

**Strengths:**

1. While the decomposition of loss into Bayes Risk and Approximation Error is standard, and the use of an "intrinsic space" is inspired by prior work, the formulation of "Intrinsic Entropy" as a central concept to bridge context length and loss is novel.
2. The idea that there is a fundamental trade-off leading to an optimal context length is a significant and non-obvious insight that challenges the simplistic view that "more context is always better."
3. The paper is generally well-structured and clear.

**Weaknesses:**

My main concern is that existing work [1] has pointed out that PPL cannot serve as a standard for evaluating long-text performance. Therefore, experimental results on more diverse long-text benchmarks are necessary, such as RULER, HELMET, and Longbench v2.

[1]Fang L, Wang Y, Liu Z, et al. What is Wrong with Perplexity for Long-context Language Modeling?[J]. arXiv preprint arXiv:2410.23771, 2024.

**Questions:**

See Weaknesses.

---

> ### Author Response · Authors · 2025-11-21
>
> We sincerely thank the reviewer for recognizing the novel idea of Intrinsic Entropy, the important observation of the trade-off behind the optimal context length, and the clarity of presentation. We also recognize down-stream tasks are important, and **we further conduct analysis for Intrinsic Entropy and optimal context length on downstream tasks, showing that**:
> 1. For down-stream tasks, there still exists an optimal context length. Moreover, this `optimal context length’ would be affected by the tasks’ different dependency on context length.
> 2. Intrinsic Entropy can still represent performance with some down-stream QA tasks.
>
>
> ## `RULER-QA1 dataset:Optimal context length and Intrinsic Entropy on downstream tasks`
>
> To study whether “Optimal Context Length” also exists for down-stream tasks, we study Llama-3.1-8B’s performance on a modified RULER-QA1[1] dataset, as added in **Appendix A in our revised paper**.
>
> The RULER-QA1 dataset is constructed by extending standard document-QA tasks through inserting the “golden paragraph” (paragraphs containing answers) uniformly into other paragraphs from the same SQuAD dataset. Our modified, Position-weighted-RULER-QA1 modifies the distribution of “golden paragraph” from uniform distribution into $P(x)\propto (1-x/L)^\gamma$, where $x$ is the distance (in token) of the “golden paragraph” to the end of input, and $L$ is the length (in token) of the concatenated sample, gamma is a hyper-parameter to control how close the golden paragraphs are distributed to end of input. (larger $\gamma$ means less long-context dependency, while $\gamma=0$ is uniform distribution).
>
> From **Appendix A Figure 9 in the revised paper**, we have these observations:
> 1. For any $\gamma$, when increasing model’s context length, the QA accuracy first improves mainly because the model can see more “golden paragraphs” (i.e. Bayes Risk decrements faster than Approximation Loss increment); then it drops because the model becomes unable to handle longer contexts (i.e. increment in Approximation Loss surpasses the decrement of Bayes Risk). Therefore, there is still an optimal context length, for certain down-stream tasks.
> 2. For smaller $\gamma$ (i.e. the task depends more on tokens farther away), the optimal context length is larger. This aligns with the intuition that, for applications that typically require larger context length, the “optimal context length” would be larger.
> 3. When increasing context length, before QA accuracy starts to decrease (i.e. model deviates from Bayes Model), the Intrinsic Entropy still increases along with QA accuracy.
>
> In this experiment, we are not training a new Language Model on this task (because training a model with moderate document QA accuracy on up-to 8k context length, like Llama-3.1-8B, would require too much computation), but we believe these results can provide evidence and particular insight to why and how optimal context length exists for down-stream tasks. That is:
>
> The final loss can be decomposed into Bayes Risk and Approximation Loss. Longer context would decrease Bayes Risk, while increase Approximation Loss since the model is less capable of obtaining key information and conducting reasoning in long input contexts. For tasks that require larger context lengths to solve (e.g. in our modified Ruler-QA1 dataset with smaller $\gamma$), the Bayes Risk decreases slowlier with context length, and hence the optimal context length is longer.
>
> ## `Synthetic Dataset resembling two-needles-in-haystack task: optimal context length on "synthetic" retrieval task`
> Our synthetic dataset proposed in Figure 5 in our original paper actually resembles the two-needle-in-haystack task. We train an MLP with fixed training dataset size provided with different visible context length on a similar synthetic task. Our results shown in **Appendix F, Figure 12 in the revised paper** also show that there exists an optimal context length that increases with training dataset size, thus proving the effectiveness of the optimal context length observation on tasks more similar to down-stream tasks.
>
> **Overall**, we hope our further analysis and experiments on downstream QA tasks could be helpful for your evaluation. Thanks again for the valuable insights.
>
> **P.S.**: We have uploaded our revised paper, where we use $\\color{blue}{\\text{blue texts}}$ to highlight updates.
>
> **Reference**:
>
> [1] What is Wrong with Perplexity for Long-context Language Modeling? COLM 2024 https://arxiv.org/abs/2404.06654

---

> > ### Comment · Reviewer_8bxz · 2025-11-26
> >
> > Thank you for the authors’ response. I’m glad to see that results on additional downstream tasks have been included. Based on my experience, downstream task evaluation is very important, and the downstream evaluations in this paper are relatively limited. In addition, as the model size increases, it may become more difficult to observe the turning point at which performance begins to decline. I am also curious whether performance would rise again as the amount of data increases.

---

> ### Author Response · Authors · 2025-11-28
>
> Thank you for further comments. We have further conducted more experiments on subsets (including qa1(document qa), vt(variable tracking), fwe(frequent word extraction) and cwe(common words extraction)) of the RULER Benchmark[1] with the Qwen3 series models (non-thinking mode), and **our results are updated with $\\color{blue}{\\text{blue texts}}$ in Appendix A.3**.
>
> ### `More experiments on subtasks of RULER Benchmark`
>
> We carried out experiments with the **Qwen-3 series models on more subsets of RULER Benchmark**, out results are shown in **Appendix A.3 in the revised paper**.
>
> As shown in **Appendix A.3 Figure A.3-1**, on **document qa, frequent word extraction and common words extraction**, we observe clear optimal visible context length for all models test. Approximation Loss of these models on these tasks begin to increase when increasing context length, surpassing the decrement of Bayes Risk, thus leads to an optimal context length. This implies that these models are yet to achieve a $0$ (or neglectable) Approximation Loss on these tasks. This implies that: our models tested become more ineffective to handle an increasing context length of this task given long context, though some can achieve as high as $90\%$ accuracy on certain task.
>
> For **variable tracking**, current models can already handle it very well (i.e. Qwen3-14B models achieve almost $100\%$ accuracy rate for a visible context length of 8k on the max context length of 8k task), we observe that accuracy often increases with visible context length. This can also be explained from the Bayes Risk - Approximation Loss decomposition perspective: the models can almost perfectly solve this task, the Approximation Loss is very small, and the loss is dominated by Bayes Risk along different context lengths, hence performance often improves with respect to visible context length.
>
> **These further experiments further prove** the existence of optimal context length for these non-saturating tasks, and show the validity of our analysis from the Bayes Risk - Approximation Loss perspective. Also, our work provides the perspective that, LLMs achieve $90\%$ accuracy on certain tasks do not mean it may not get disracted by the more context provided; this could potentially be insight for future benchmarks targeting to evaluate the long context performance of LLMs. In application scenarios[2,3] where LLMs are used as coding agents, etc., the tasks would be more difficult and more similar to the **doc qa** tasks, where we indeed observe an optimal context length.
>
> ### `Analysis with respect to model size`
>
> For model size: it is indeed the case that **stronger models tend to perform better on longer contexts**. For example, for the CWE task shown in **Figure A.3-1 in Appendix A.3**, the Qwen3-4B model shows an optimal visible context length of around $0.4\*MaxCtl$, while for Qwen3-8B model such optimal context length is around $0.8\*MaxCtl$. This is because larger and stronger models tend to have smaller Approximation Loss, hence the balance between Bayes Risk and Approximation Loss would result in a longer optimal context length.
>
> Given that said, though **optimal context length is harder to observe for larger model, it is easier to observe for long context tasks**. As shown in **Figure A.3-2 in Appendix A.3**, Qwen3-4B model shows an increasing performance with respect to visible context length (i.e. optimal context length hard to observe) when the task has max context length of $8k$, but it shows an increasing-then-decreasing (i.e. optimal context length easy to observe) performance w.r.t. visible context length when task has max context length of $16k$. This implies that a task requiring longer context to solve would make optimal context length easier to observe: this is consistent with our findings in **Appendix A.1 and A.2 in our first rebuttal**.
>
> For many real application scenarios such as coding[2,3], understanding long context and performing thorough reasoning is needed, and current state-of-the-art LLMs' performance are still far from saturation. Therefore, our analysis on optimal context length can be insightful for these scenarios[2,3].
>
> **Overall**, our further experiments have shown the existence of optimal context length for long-context tasks, and further shows the validity of our analysis. We would like to thank the reviewer for the insightful comments that have helped to improve the soundness of our work.
>
> Reference:
>
> [1] RULER: What's the Real Context Size of Your Long-Context Language Models?, COLM 2024, https://openreview.net/forum?id=kIoBbc76Sy.
>
> [2] SWE-bench: Can Language Models Resolve Real-World GitHub Issues?, ICLR 2024, https://openreview.net/forum?id=VTF8yNQM66.
>
> [3] Introducing SWE-bench Verified, Openai blog, https://openai.com/index/introducing-swe-bench-verified/.

---

### Author Response · Authors · 2025-12-03
**Overall Comment by Authors**

We would like to thank all reviewers and AC for their efforts made for reviewing our work. In this overall comment, we would like to (1) summarize our work, (2) summarize Pros, Cons and Questions from every reviewer, our rebuttal, and **each reviewer’s response (all responses are before the incident)**. We hope this global comment would be helpful for further evaluation.
### `Summary of our work`

**In our work**, we study the performance of Large Language Models (LLMs) performance with respect to context length. We mainly focused on cross entropy loss of certain context length, and focused on decomposing the loss into Bayes Risk and Approximation Loss. When increasing context length, Bayes Risk decreases (because more information is seen) while Approximation Loss generally increases (because LLMs perform worse on longer contexts). (1) For Bayes Risk, we propose to use Intrinsic Entropy to explain how Bayes Risk changes; (2) for the total loss, we propose the existence of optimal context length caused by a balance between Bayes Risk and Approximation Loss. These statements are validated on Natural Language dataset and Synthetic Dataset as well.

**In our rebuttal and our revised paper**, we further conduct experiments focusing on down-stream tasks, represented by the RULER subsets. We show that for these kinds of down-stream tasks such as document QA, our proposals of (1) using Intrinsic Entropy measured to measure intrinsic information; and (2) the analysis of context length from Bayes Risk and Approximation Loss, and the existence of an optimal context length in most non-saturated cases are still valid. Also, we conduct further experiments with more model series, with models from the Qwen-3 series and the Recurrent Gemma series.

---

> ### Author Response · Authors · 2025-12-03
>
> ###  `Summary of discussions with Reviewer vDBt`
>
> **Reviewer proposed Pros**: Reviewer vDBt thought highly of our non-trival observation of intrinsic entropy and optimal context length, and our consistent throy. Also the reviewer thinks our experiments are supportive for our theory assumptions.
>
> **Reviewer proposed Cons and Questions**: (1)Reviewer vDBt thought our original paper lacks downstream tasks, and also (2) perhaps in some real-world scenarios that require really long contexts, perhaps we have to use longer context length even if there exists some certain ‘optimal context length’.
>
> **Author-Reviewer discussions**: (Similar to rebuttal with Reviewer 8bxz) we conducted further thorough experiments on the RULER dataset as downstream tasks, results of which are shown in Appendix A. Also, we argued that for scenarios requiring long contexts, the ‘optimal context length’ itself would be long. The reviewer's **score hadn’t changed as 10 (conf: 4) during the discussion phase**.
>
> ###  `Summary of discussions with Reviewer ULtS`
> **Reviewer proposed Pros**: Reviewer ULtS commented on (1) our novel introduction of Intrinsic Entropy to link information measures to model-internal representation, Bayes Risk and Approximation Loss, and (2) our thorough empirical discoveries on NLP datasets and synthetic datasets.
>
> **Reviewer proposed Cons and Questions**: (1) we assumed the existence of a stable intrinsic space for language modeling; (2) experiments in original paper are mainly on Llama-3 models, not with other kinds of model series or architectures; and (3) Intrinsic entropy could have heavy computational overhead; and questions (1) more general definition of intrinsic entropy; (2) study down-stream task performances, and (3) conduct experiments for more models.
>
> **Author-Reviewer discussions**: We further conducted experiments with more model architectures and proved the validity of our defined Intrinsic Entropy across multiple models in Appendix G of revised paper. Also, we conducted more analysis of Intrinsic Entropy and optimal context length in more down-stream tasks. Also, Intrinsic Space is a widely adopted concept for previous work. The reviewer thought our response had clarified the concerns, and **updated the score from 6 to 8 with confidence 3 (Nov. 26, 1:00 am UTC)**.

---

> ### Author Response · Authors · 2025-12-03
>
> ###  `Summary of discussions with Reviewer 8bxz`
> **Reviewer proposed Pros**: Reviewer 8bxz commented on (1) our idea of Intrinsic Entropy as novel, (2) the discovery for the trade-off behind the optimal context length as important, and (3) thought our paper is well-structured and clear.
>
> **Reviewer proposed Cons and Questions**: Reviewer 8bxz was concerned that in our original paper, we were mainly focusing on Cross Entropy loss, lacking experiments on down-stream tasks.
>
> **Author-Reviewer discussions**: We conducted experiments on RULER qa1 subtask, showing the validity of the intrinsic entropy and the context length. After **reviewer 8bxz expressed gladness** to see our first-round rebuttal result and expressed interests about more experiments and models with various size, we further conducted more thorough experiments on more subtasks (Doc-qa, common word extraction, variable tracking, frequent word extraction) of RULER with the Qwen3 series models of different size, showing the concept of optimal context length: the existence of optimal context length is harder to observe for larger models, but easier to observe for tasks that require longer contexts. The reviewer’s score **hadn’t changed as 4 with conf 4 until the discussion was closed** due to the incident. We conducted more experiments and made another reply to the reviewer's further comment, **but unfortunately due to the incident we could not have further discussions** with the reviewer.
>
>
>
> ### `Summary of discussions with Reviewer Mnty`
>
> **Reviewer proposed Pros**: Reviewer Mnty thought (1) the decomposition of loss and the existence of optimal context length is very interesting, (2) the experiments are thorough, and (3) thought the discovered relationship between cross entropy and intrinsic entropy is novel.
>
> **Reviewer proposed Cons and Questions**: Reviewer Mnty thought (1) the math lacks rigor, for example we are taking derivative of Bayes Risk to context length $l$ and $l$ is an integer, and thought we are implying equaling states with volume divided by dimensions when calculating intrinsic entropy, and (2) the presentation is bad, with figures of non informative titles, and Reviewer Mnty was confused and not fully understood how our experiments were conducted.
>
> **Author-Reviewer discussions**: (1) We modified our paper, added certain comments and reference to figures; (2) we claimed that taking derivative over an integer like context length $l$ or number parameter $N$ is widely adopted in previous work and we followed previous work, while we also add more related comments for math rigor in our revised paper; and (3) we claimed that used such notation for calculating intrinsic entropy is for emphasizing the homogeneous reference measure concept, which is equivalent to an assumption in the Gaussian differential entropy estimation.
>
> Reviewer Mnty accepted our explanation with respect to context length and our calculation of intrinsic entropy, and **updated confidence from 3 to 2, kept overall score as 2, and increased soundness score** (on Nov.23, UTC). Reviewer Mnty also expressed other concerns like whether it is overfitting to use the same metric in training and evaluation, and some other figure presentation issues in the second-round comment, which we replied but unfortunately had no chances for further discussions due to the incident.
>
> **Overall**, the discussions are constructive and have helped improve our work. We are grateful to reviewers and AC for their efforts in reviewing our work.

---

### Meta-Review · Area_Chair_Y8yx · 2026-01-04

**Summary:**

This paper introduces a novel theoretical framework using "Intrinsic Entropy" to explain the impact of context length on Language Model performance, adding another dimension (context length) into scaling law. It decomposes the cross entropy loss into Bayes Risk (irreducible error, decreases with context length) and Approximation Loss (modeling error, increases with context), predicting an "optimal context length" for a given training budget. Empirically, experiments using LLaMA-3.1 (8B and 70B) on OpenWebText subsets and a synthetic parity dataset confirm the theory. The findings imply that excessively extending context windows can degrade performance when data are insufficient, emphasizing the need for joint optimization of context length, data size, and model capacity. The paper offers a fresh and scientifically grounded perspective on context length scaling. The rebuttal successfully addressed the primary criticism regarding the lack of downstream task evaluation. The result is a solid contribution to the understanding of LLM scaling laws.

Although this paper receives mixed scores (4, 2, 10, 6), with the rebuttal, I think most concerns should be addressed, which the 4, 2 reviewers both acknowledged at least partially. The authors should ensure all additional experiments on RULER and the Qwen/Gemma models are integrated seamlessly into the main text or appendices of the camera-ready version, and that the final presentation polish requested by Reviewer Mnty is maintained.

**Reviewer Concerns:**

There is only one common, essential concern, i.e., lack of downstream/functional evaluation (Reviewers 8bxz, vDBt, ULtS): The original paper focused heavily on perplexity/cross-entropy loss. Reviewers argued that PPL is insufficient for long-context evaluation and requested functional metrics (e.g., RAG accuracy, F1 scores) on benchmarks like RULER. The authors added extensive experiments on the RULER benchmark (specifically QA1, Document QA, Variable Tracking, etc.) in the revised Appendix. They demonstrated that the concept of "optimal context length" holds true for these downstream tasks (where accuracy initially rises then falls), validating their theoretical framework beyond simple loss metrics. Reviewers 8bxz and ULtS explicitly acknowledged these additions as helpful.

**Reviewer Scores:**

I believe reviewer 8bxz would increase their score from 4 to 6. Reviewer Mnty (overall score 2) explicitly stated that they wouldn't increase the overall score, but has increased the soundness score and lowered the confidence. They also acknowledged that this is a potentially exciting paper, but the main issue is the presentation.

---

### Decision · Program_Chairs · 2026-01-26

Accept (Oral)